# UMP-Net: Uncertainty-Aware Mixture of Prompts Network for Efficient Instruction Tuning

**Fatemeh Daneshfar**                                          *f.daneshfar@uok.ac.ir*
*Department of Computer Engineering, University of Kurdistan, Sanandaj, IRAN*

**Abdulhady Abas Abdullah**                                    *abdulhady.abas@ukh.edu.krd*
*Artificial Intelligence and Innovation Centre, University of Kurdistan Hewler, Erbil, Iraq*

**Moloud Abdar** *                                            *m.abdar1987@gmail.com*
*CHIRP, Child Health Research Centre, The University of Queensland, Brisbane, Australia*

**Pietro Liò**                                                *pl219@cam.ac.uk*
*Department of Computer Science and Technology, University of Cambridge, Cambridge, UK*

**Reviewed on OpenReview:** *https://openreview.net/forum?id=EehtvgNXAl*

## Abstract

Instruction tuning has greatly improved how large language models (LLMs) respond to human-like instructions. However, fully fine-tuning these models is still computationally demanding, and many existing parameter-efficient methods fall short, particularly when it comes to uncertainty estimation and working effectively across different modalities. To address this, we introduce UMP-Net (Uncertainty-Aware Mixture of Prompts Network), a new approach designed to enhance the ability of LLaMA to follow instructions. UMP-Net combines a novel mixture of prompts (MoPs) technique with Latent Noise Prompting, KNN-based Heterogeneous Clustering, and Conformal Predictions to select the most reliable prompts dynamically while accounting for uncertainty. In addition, it features a CLIP-based multi-modal architecture to streamline vision-language integration. We evaluated UMP-Net on a range of benchmarks including ScienceQA, COCO Caption, and various zero-shot multi-modal tasks. The results show a strong performance: an average accuracy of 88.41% on ScienceQA and a CIDEr score of 158.3 on COCO Caption, surpassing models such as LLaVA, LLaMA-Adapter, and LLaMA-Excitor. These findings suggest that UMP-Net offers both improved multi-modal capability and computational efficiency. Further ablations demonstrate UMP-Net's conformal prediction module provides robust uncertainty estimates under noise and domain shifts, outperforming Bayesian alternatives in coverage guarantees with minimal overhead. The code of our proposed model is available here: https://github.com/abdulhadyabas2/UMP-NetUncertainty.

## 1 Introduction

Instruction tuning is rapidly an important way of enhancing the ability of large language models (LLMs) to respond to and obey human instructions in a broad task library Ouyang et al. (2022); Wei et al. (2022). Early successes with models like FLAN Wei et al. (2022) and InstructGPT Ouyang et al. (2022) highlighted how fine-tuning pre-trained LLMs using instruction datasets could significantly boost their zero-shot and few-shot performance. With these gains, the vast majority of these methods require full model fine-tuning, a process that not only is expensive in terms of resources but also is unfeasible when dealing with very large models like LLaMA Touvron et al. (2023), which have billions of parameters. Besides, there is an added

---

*Correspondence to Moloud Abdar <m.abdar1987@gmail.com>

complexity of multi-modal large language models (MMLMs). Visual input with textual input can require further extensive pre-training or fine-tuning to train, further increasing the already high computational costs (Liu et al., 2023b; Li et al., 2023a).

Researchers have come up with parameter-efficient fine-tuning (PEFT) methods, including LoRA Hu et al. (2021) and prompt tuning Lester et al. (2021), that fine-tune only a small fraction of model parameters, without changing the underlying language model. Nevertheless, these methods are frequently unable to accomplish zero-shot generalization on a variety of tasks, especially in multi-modal contexts, where visual and text data representations are vital to merge. Our work is motivated by the fact that the current PEFT techniques have two significant weaknesses: they can process multi-modal inputs only to a limited degree and do not provide any powerful tools to quantify and control the uncertainty of prediction. The models in practice, e.g. medical diagnostics or autonomous systems, need to take ambiguous or noisy inputs across modalities and give reliable outputs with quantifiable confidence. The current solutions such as Flamingo Alayrac et al. (2022) and LLaVA Liu et al. (2023b) are based on the need to use large-scale datasets to perform vision-language alignment, which are computationally intensive and cannot work in the resource-limited setting. The second major limitation is that there are no tools to detect and deal with uncertainty in model prediction. This is of great essence especially when handling ambiguous or noisy input because a clear direction is highly required. Such difficulties become even more evident with multi-modal applications in which the lack of adaptability at any point in time to a particular task and modality is combined with the lack of uncertainty-aware components to complicate performance and reliability.

In this paper, we introduce the UMP-Net (Uncertainty-Aware Mixture of Prompts Network) that can be used to overcome the constraints of current instruction-tuned and multi-modal systems. UMP-Net is an integration of uncertainty-aware prompt tuning with an effective approach to multi-modal adaptation. Its key idea is a blend of a mix of prompts (MoPs) mechanism, a union of Latent Noise Prompting, KNN-based Heterogeneous Clustering (HeteroGraphPrompt), and Cluster-Wise Uncertainty Estimation (CUE) to dynamically support prompts to the LLaMA model. The system uses Conformal Predictions to enhance reliability to enable it to quantify uncertainty both at prompt and cluster levels and use the measures to inform the selection process. At the multi-modal interface, UMP-Net applies CLIP-based embeddings Radford et al. (2021) to incorporate visual data, which allows to perform effective cross-modal reasoning without expensive pre-training. Not only will this enhance the ability of LLaMA to adhere to instructions in a language-only and multi-modal context, but also, it maintains computational requirements low, which is why it can be readily used in low-resource environments. To measure the effectiveness of our proposed UMP-Net, we have made a comparative analysis with other existing models, LLaMA-Adapter Zhang et al. (2024) and LLaMA-Excitor Zou et al. (2024), in various tasks with a combination of visual and textual inputs. This comparison can be seen in Figure 1 and shows that UMP-Net is better in tasks that include the recognition of solution concentrations, botanical features, the description of medical specialties and the creation of functional code.

The contributions made by this paper are three. (1) We present UMP-Net, a parameter-efficient model that integrates multi-modal-adaptation with uncertainty-oblivious prompt tuning. It achieves state-of-the-art text-only and vision-language benchmark results. (2) Our suggestion is a clustering and uncertainty new estimation pipeline based on the use of KNN-based prompt categorization and Conformal Predictions. It helps to increase timely reliability and reduction of redundancy. (3) We test the performance of UMP-Net by extensive benchmark testing Systems like ScienceQA Lu et al. (2022a), COCO Caption Chen et al. (2015), and a spectrum of zero-shot multi-modal tasks. UMP-Net is steadily doing so across these settings performs better than the top models such as LLaVA Liu et al. (2023b), LLaMA-Adapter Zhang et al. (2024), and LLaMA-Excitor (Zou et al., 2024).

## 2    Proposed Method

This section introduces a new framework, UMP-Net, which has been created to augment the LLaMA model by improving its performance using a learnable adaptation prompt, which combines mixture of prompts strategy. We combine Bayesian reasoning, Conformal Prediction and heterogeneous clustering using KNN to

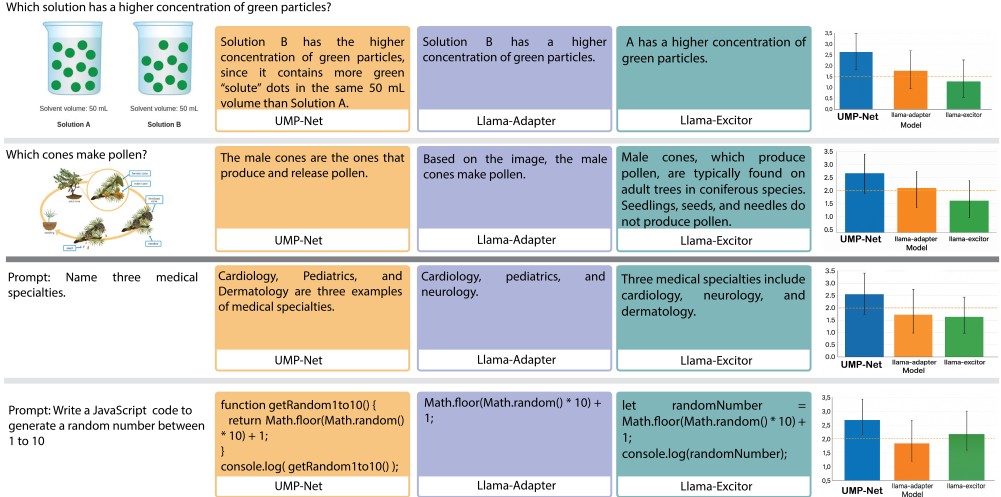

Figure 1: Comparison of UMP-Net, LLaMA-Adapter Zhang et al. (2024) and LLaMA-Excitor Zou et al. (2024) on four mixed visual-text tasks: determining the concentration of solutions, identifying pollen-producing cones, enumerating medical specialties, and writing JavaScript code. Right bar charts indicate the better mean scores in human evaluation through standard deviation error bars at UMP-Net.

develop a more robust and uncertainty-sensitive prompting procedure. This combination will enable UMP-Net to respond dynamically to customizing prompts and have a high level of reliability of its predictions.

## 2.1 Overview of UMP-Net

UMP-Net is a modular architecture, which is structured to dynamically create and assign weights to prompts in relation to their corresponding scores of uncertainty. There are three fundamental elements of the model as shown in Figure 2. Latent Noise Promoting with MoPs strategy (1), KNN based Heterogeneous Clustering based on prompt selection and aggregation (2), and conformal predictions to estimate uncertainty among prompt candidates (3). Attention Gates and Softmax layers connect these modules and help the system to calculate one effective prompt, which is reliable and easily adapted to the requirements of the LLaMA model.

## 2.2 Latent Noise Prompting with MoPs

The Latent Noise Prompting module is an essential feature of UMP-Net that is supposed to cause controlled variability to the prompt generation process to adapt LLaMA. The purpose of this module is to allow dynamic prompt generation that can respond to different tasks and inputs and overcome the drawbacks of traditional prompt tuning methods that are problematic with task-specific generalization. This module adds to the capability of the model to explore a larger repertoire of prompt representations to increase robustness and flexibility both in a language-only and multi-modal context. The module starts with sampling latent noise $Z$ as a standard Gaussian noise $N(0, I)$, with $I$ denoting the identity matrix, to make the noise be isotropic with zero mean and unity variance. The number of dimensions of $Z$ is defined as $d_z$ i.e. the dimension of the latent space which is often equal to the input embedding size of the LLaMA model.

The latent noises are sampled as the $Z \in \mathbb{R}^{d_z}$ and fed into a Multi-layer Perceptron (MLP) to produce an MoP, denoted as $P_{1:n}$, where $n$ is the number of prompts in the mixture. The different semantic and syntactic properties of the possible inputs are represented by each prompt $P_i \in \mathbb{R}^{d_p}$ (in which the embedding dimension of the prompt is denoted by $d_p$). The MLP, parameterized by weights $W^{(l)}$ and biases $b^{(l)}$ across $L$ layers, transforms the latent noise as follows:

$$H^{(l)} = \sigma(W^{(l)}H^{(l-1)} + b^{(l)}), \quad l = 1, 2, \ldots, L, \tag{1}$$

where $H^{(0)} = Z$, $H^{(L)} = P_{1:n}$, and $\sigma$ is a non-linear activation function (ReLU). The output of the resulting $P_{1:n}$ becomes the input of the next modules, including Conformal Prediction and Heterogeneous Clustering, to further select and weight prompts.

## 2.3 Heterogeneous Clustering by KNN

In order to narrow the focus of prompt selection and improve the adaptability of the UMP-Net to LLaMA, we present a heterogeneous clustering technique which uses KNN. The reason behind this module is that it is essential to deal with a variety of input modalities (textual, visual and cross-modal) and grouping prompts in sets according to their feature representations. This clustering also makes sure that prompts are specific to particular types of tasks, which makes the model more effective and strong in terms of managing multi-modal inputs. The MoPs $P_{1:n}$ are arranged into different groups (textual, visual, and cross-modal) into this module depending on the presence of their features representations, which makes it possible to treat various modalities of inputs effectively and provides the ability to be more robust when handling various tasks.

Our KNN algorithm will be used to find prompts of structural similarity to cluster them into $K$ clusters $C_{1:K}$, each of which will have $m_k + 1$ prompts. In this notation $m_k$ is the count of prompts in cluster $k$ and the $+1$ is due to a representative prompt or centroid prompt. The KNN clustering is implemented in the following way:

$$\text{Distance}(P_i, P_j) = \|P_i - P_j\|_2, \tag{2}$$

where $\| \cdot \|_2$ is the Euclidean distance. Prompts are classified into groups $C_k$ based on a set of $k$ nearest neighbors of $\text{Distance}(P_i, P_j)$ for each prompt $P_i$. This strategy has various advantages, especially where there is a huge amount of prompts to be handled so as to eliminate redundancy, boost performance and increase the uncertainty quantification in downstream modules such as Conformal Predictions. KNN-based clustering groups prompts into functional-specific coherent clusters, with each cluster modality specific: Visual Prompts (V-Prompts Cluster, $C_k^V$): Contains prompts that are specialized in processing visual information, i.e. object recognition, spatial reasoning, or image understanding. These prompts are grouped together by similarities of visual features, e.g., vision transformer embeddings. Textual Prompts (T-Prompts Cluster, $C_k^T$): Groups prompts are aimed at textual reasoning, e.g. sentence embeddings, text completion, or semantic parsing. Clustering is based on a language features similarities, which is obtained through an encoder of a language model. Unified Cross-Modal Prompts (VL-Prompts, $C_k^{VL}$): A combination of prompts with tasks requiring modalities in both visual and linguistic modalities, e.g. visual question answering or image captioning. These prompts are grouped according to joint embeddings which are the combination of visual and textual entities.

Furthermore, the absence of prompts clustering may result in redundancy or conflicting prompts and therefore, to operate UMP-Net effectively, it is challenging to calibrate and fuse a large set of prompts. The KNN-based clustering algorithm can be used to solve this problem by clustering structurally similar prompts into operational units and eliminating redundancy and making prompt management simpler. The prompts are sorted into separate clusters, e.g., $C_k^V$, $C_k^T$, and $C_k^{VL}$, and the overlap between them is reduced to a minimum to make each cluster fulfill its own. The modality-specific clustering used has $K = 3$, which allows splitting prompts into only three clusters.

## 2.4 Conformal Predictions for Uncertainty Quantification

To measure uncertainty in the UMP-Net, we use Conformal Predictions, a distribution free statistical model which gives develops good uncertainty measures of model outputs. The reason of having the knowledge of uncertainty by using Conformal Predictions is that it guarantees accurate and timely selection under ambiguous or noisy input, which is essential in robust performance in multi-moded tasks and high-stakes problems such as medical diagnostics. This module reduces the probability of using inappropriate prompts that will result in confident and yet erroneous results and make LLaMA more reliable and flexible when dealing with a wide range of tasks by measuring how uncertain the predictions of each prompt are. It evaluates the reliability of every prompt in the set $P_{1:n}$ by calculating nonconformity scores, the measure of the adherence of a given prompt to the anticipated output given a particular input. The scores are then used to obtain the confidence level and, therefore, the most trusted prompts are selected to adapt LLaMA. Given a prompt $P_i$, we calculate a nonconformity score $S(P_i, x, y)$ with respect to the input $x$ and the associated

label or output $y$, with $x$ being the input to the task (e.g. text, image or other multimodal data), and $y$ the predicted or desired output. The nonconformity scores are computed differently for each prompt type (visual, textual, or cross-modal) within their respective clusters, as detailed below.

**Visual Prompts (V-Prompts).** The nonconformity score of visual prompt $P_i$ represents the similarity of the input image with the visual features that the visual cluster $P_i$ anticipates. We define the V-Prompt nonconformity score as:

$$S(P_i, x, y) = \|f(x) - g_i(y)\|_2^2, \tag{3}$$

where $f(x) \in \mathbb{R}^{d_v}$ is the visual feature representation of the input image $x$, as computed by a pre-trained vision model (such as a convolutional neural network, or vision transformer). $g_i(y) \in \mathbb{R}^{d_v}$ denotes the embedding of the $i$-th visual prompt $P_i$ within the cluster, that is an expression of the output $y$ (e.g. a predicted class or description) into the visual feature space.

The intuition of this formulation is that the combination of the Euclidean distances of this prompt will capture the nonconformity measure of the collective similarity of the input image to the visual features of the prompt that were supposed to be seen. A smaller $S(P_i, x, y)$ will be an increased conformity (i.e. the input fits well into the prompt), which means an increased uncertainty.

**Textual Prompts (T-Prompts).** The nonconformity score in the case of the cluster of textual prompts, which uses the linguistic input to predict, is the negative log-likelihood of the label $y$ given the input $x$. The prompt-level nonconformity score is:

$$S(P_i, x, y) = -\log P_i(y|x), \tag{4}$$

where $P_i(y|x)$ is the likelihood of the label $y$ returning the input $x$, which is forecasted by the $i$-th textual prompt $P_i$ in the cluster.

This cumulative score is a measure of overall confidence of the textual prompts in the cluster. As conformity (i.e. less uncertainty) is desired, a higher $S(P_i, x, y)$ means that the probability that the predicted label $y$ matches the input $x$ is reduced. The probabilistic nature of language models is used in this formulation to make sure that uncertainty is measured in predictive confidence.

**Unified Cross-Modal Prompts (VL-Prompts).** In the case of cross-modal prompts where modalities are dependent on each other (text and visual), we determine a weighted hybrid nonconformity rating by weighting the two domains. The cluster-level nonconformity score is given by:

$$S(P_i, x, y) = \left[\lambda \|f(x) - g_i(y)\|_2^2 - (1-\lambda) \log P_i(y|x)\right], \tag{5}$$

where $\lambda \in [0, 1]$ is a hyperparameter that balances the contributions of the visual ($\|f(x) - g_i(y)\|_2^2$) and textual ($-\log P_i(y|x)$) components. Moreover, $f(x)$, $g_i(y)$, and $P_i(y|x)$ are defined as in the visual and textual cases, respectively.

This expression leads the score of nonconformity to portray the view of the cluster as a collective judgment as far as modalities are concerned. The parameter $\lambda$ can be adjusted according to the task need whereby there is a flexibility of focusing on either the visual or the text information. The smaller $S(P_i, x, y)$, the greater the conformity and the less uncertainty, which allows UMP-Net to adapt LLaMA to multimodal inputs successfully. The score of all prompts nonconformity is used to select the most confident prompt in each cluster, $P_{\text{best}}$, to use in LLaMA adaptation as explained in the Attention Gate and Weighted Prompt Creation module.

## 2.5 Attention Gate and Weighted Prompt Creation

The most promising prompts in each cluster are forwarded to an Attention Gate which weighs each prompt dynamically depending on its relevance. Attention Gate uses Softmax to normalize attention scores, which results in weighted prompt $P_{\text{weighted}}$. The weighting process is guided by:

$$P_{\text{weighted}} = \sum_{k=1}^{K} \alpha_k P_{\text{best},k}, \tag{6}$$

where $\alpha_k$ is the weight of attention of prompt $P_{\text{best},k}$, learn with the training time. The weighted prompt of outputs is then chosen to be adapted by LLaMA.

The last weighted prompt is incorporated into the LLaMA model, which allows it to be open to inputs. UMP-Net enhances LLaMA on generating consistent and contextually suitable replies to specific matters, especially when there is minimal or noisy data (see Figure 2). Algorithms The UMP-Net pipeline may be summarized as follows (Algorithms 1).

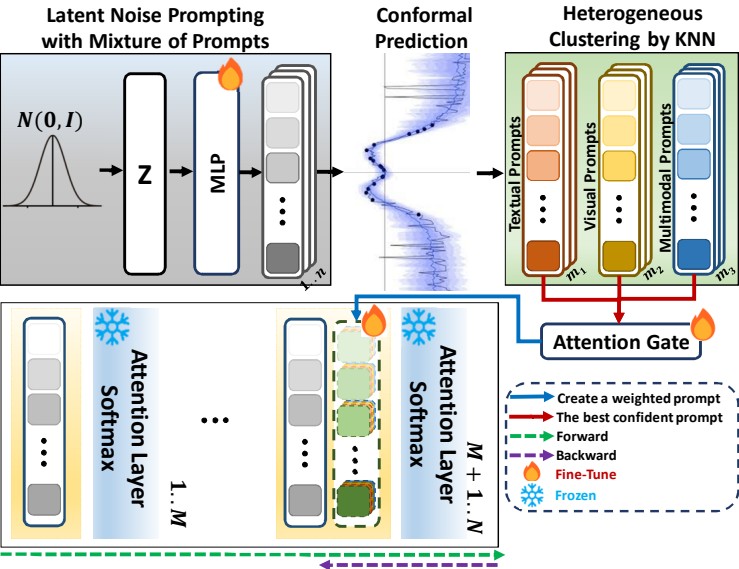

Figure 2: Figure illustrates UMP-Net architecture when adapting to LLaMA. The first step is Latent Noise Prompting, in which noise in a normal distribution $N(0, I)$ is fed to an MLP to produce prompt initials. These prompts are Conformally Predicted to determine their uncertainty, and then Heterogeneous Clustering is used with KNN to determine them as textual, visual, and multi-modal clusters. Each cluster is then picked by an Attention Gate through a Softmax Layer and it involves picking the best confident prompt to generate a weighted prompt. LLaMA is finally modified to incorporate the final weighted prompt in order to achieve better instruction-following and frozen layers make it efficient.

## 2.6 Multi-modal Architecture

Multi-modal architecture of UMP-Net increases the capacity of the system to handle different modalities of input by embedding images at different points in the pipeline as demonstrated in Figure 3. This module rationale is that it will facilitate a smooth integration of visual and textual information to overcome the issue of matching multi-modal inputs when there are constraints on resources in which pre-training on a large scale is impractical. This architecture uses CLIP model Radford et al. (2021) to project multimodal features and integrates image embeddings into the prompt embeddings following a clustering and distributed across all the attention layers, making sure that it is strongly integrated multimodally. The architecture incorporates the following important components and processes:

**CLIP-based Image Embedding.** The input image $x_{\text{img}}$ (e.g., the cat image in Figure 3) is processed through CLIP to extract a visual embedding:

$$e_{\text{img}} = \text{CLIP}_{\text{visual}}(x_{\text{img}}) \in \mathbb{R}^{d_c}, \tag{7}$$

where $d_c$ is the CLIP embedding dimension, aligning with the prompt embedding dimension $d_p$.

**Embedding Addition to Each Prompt.** After Heterogeneous Clustering by KNN, the image embedding $e_{\text{img}}$ is added to each prompt $P_i \in C_k$ within clusters $C_k^V$, $C_k^T$, and $C_k^{VL}$. For each prompt $P_i \in \mathbb{R}^{d_p}$, the augmented embedding is computed as:

$$P_i^{\text{aug}} = P_i + W_{\text{proj}}e_{\text{img}}, \tag{8}$$

---

**Algorithm 1** UMP-Net Algorithm for LLaMA Adaptation

---

**Require:** Input $x$ (task input), $d_p$ (latent dimension), $n$ (number of prompts), $K$ (number of clusters), $k$ (KNN neighbors), $L$ (MLP layers), $\lambda$ (cross-modal weight), pre-trained LLaMA model
**Ensure:** Weighted prompt $P_{\text{weighted}}$, predicted output $y_{\text{pred}}$

1: **1. Latent Noise Prompting**: Sample $Z \sim N(0, I)$ with dimension $d_z$
2: Process $Z$ through MLP with $L$ layers ($H^{(l)} = \sigma(W^{(l)}H^{(l-1)} + b^{(l)})$) to generate $P_{1:n} \in \mathbb{R}^{d_p}$
3: **2. Heterogeneous Clustering by KNN**:
4: Partition into $K$ clusters $C_{1:K}$ ($C_k^V$, $C_k^T$, $C_k^{VL}$)
5: **3. Conformal Predictions**:
6: **for** each $P_i \in P_{1:n}$ **do**
7:     **if** $P_i \in C_k^V$ **then**
8:         $S(P_i, x, y) = \|f(x) - g_i(y)\|_2^2$                                       ▷ Visual
9:     **else if** $P_i \in C_k^T$ **then**
10:         $S(P_i, x, y) = -\log P_i(y|x)$                                         ▷ Textual
11:     **else if** $P_i \in C_k^{VL}$ **then**
12:         $S(P_i, x, y) = \lambda\|f(x) - g_i(y)\|_2^2 - (1-\lambda)\log P_i(y|x)$     ▷ Cross-modal
13:     **end if**
14: **end for**
15: Select best prompt $P_{\text{best},k}$ per cluster with lowest $S(P_i, x, y)$
16: **4. Attention Gate**:
17: Compute $P_{\text{weighted}} = \sum_{k=1}^{K} \alpha_k P_{\text{best},k}$ through learning
18: **5. LLaMA Integration**:
19: Feed $P_{\text{weighted}}$ into LLaMA to get $y_{\text{pred}} = \text{LLaMA}(x, P_{\text{weighted}})$
20: **return** $P_{\text{weighted}}, y_{\text{pred}}$

---

where $W_{\text{proj}} \in \mathbb{R}^{d_p \times d_c}$ is a learnable projection matrix ensuring dimensional compatibility ($d_p = d_c$ after projection).

**Confidence Score Computation for Each Prompt.** Using the augmented prompts $P_i^{\text{aug}}$, we recompute the nonconformity scores as described in Section 2.4. For each prompt $P_i$, the nonconformity score $S(P_i^{\text{aug}}, x, y)$ is calculated based on its cluster type. The confidence score $\text{conf}(P_i^{\text{aug}})$ is then derived as the inverse of the nonconformity score:

$$\text{conf}(P_i^{\text{aug}}) = \frac{1}{1 + S(P_i^{\text{aug}}, x, y)}, \tag{9}$$

ensuring that lower nonconformity (higher conformity) corresponds to higher confidence.

**Selection of Best Prompts from Each Cluster.** For each cluster $C_k$, we select the prompt with the highest confidence score as the best confident prompt:

$$P_{\text{best},k} = \arg \max_{P_i^{\text{aug}} \in C_k} \text{conf}(P_i^{\text{aug}}). \tag{10}$$

This results in $K$ best prompts $P_{\text{best},1:K}$, one from each cluster.

The selected best prompts $P_{\text{best},k}$ are passed to the Attention Gate, which computes attention weights $\alpha_k$ using a softmax layer.

**Multimodal Integration**: The integration of $e_{\text{img}}$ into each prompt enhances UMP-Net's ability to handle tasks such as visual question answering (e.g., processing the cat image in Figure 3). The confidence-based selection and attention mechanism ensure that the most reliable prompts are prioritized, improving the quality of the final weighted prompt for the LLaMA adaptation.

This multi-modal architecture strengthens UMP-Net's capability to process diverse data types, leveraging CLIP's pre-trained visual representations and the systematic integration of image embeddings to optimize performance for LLaMA adaptation. This proposed method significantly improves LLaMA's robustness and adaptability, as demonstrated in subsequent experimental sections.

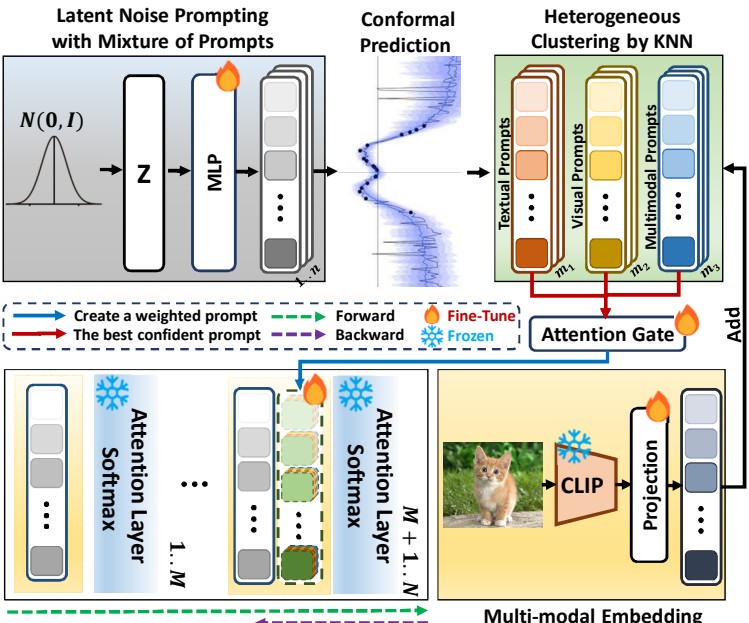

Figure 3: Schematic representation of the multi-modal architecture of UMP-Net with the focus on the combination of visual and textual embeddings to achieve greater LLaMA adaptation. One major multi-modal step is to incorporate CLIP-based images embeddings to every prompt to generate augmented multi-modal embeddings.

## 3 Experiments

### 3.1 Language Only Performance Assessment

**Experimental Setup**. Following the Stanford Alpaca Taori et al. (2023a), we employ a data set of 52K instruction-following examples for training purposes. The UMP-Net model is fine-tuned using 2 RTX 4090 GPUs over 4 epochs. We configure the training with two warmup epochs, a batch size of 8, a learning rate of 0.009, and a weight decay of 0.02. By default, we utilize the LLaMA-Adapter Zhang et al. (2024) pre-trained for version LLaMA2 7B and the foundation pre-trained LLaMA model with 8B version LLaMA3 parameters and $N = 32$ transformer layers. The prompt length is set to $d_p = 40$, and the adaptation prompts are integrated into the final $M = 30$ layers of the model. For quantitative evaluation, we compare our approach against methods trained on the same 52K instruction dataset, specifically Alpaca Taori et al. (2023a), Alpaca-LoRA Tloen (2023), LLaMA-Adapter Zhang et al. (2024) and LLaMA-Excitor Zou et al. (2024), using the widely recognized GPT-4 evaluation benchmark (Chiang et al., 2023b).

**Instruction-Following Performance Assessment**. Table 1 provides a comprehensive evaluation of the ability to follow instruction in the proposed UMP-Net model and competing approaches.

This question evaluates the ability of UMP-Net to comprehend abstract philosophical ideas, to combine retrospective views of history, and to justify the choice, evaluating the critical thinking and linguistic understanding abilities. In Table 1, the answers provided by different models to the question of defining intelligence by the different philosophical approaches are compared. The advantage of UMP-Net is that it provides a holistic definition of intelligence which combines cognitive, practical, moral, and emotional aspects relying on the philosophies of Plato, Aristotle, and Confucius and provides a balanced preference. Conversely, LLaMA7B offers a very specific definition of problem-solving, which is not philosophical. Both Alpaca-LoRA and LLaMA-Adapter Zhang et al. (2024) authors talk about Plato and Aristotle without using any elements of modern or emotionality, which restrict the scope of their responses. LLaMA-Excitor Zou et al. (2024) discusses various theorists but lacks a synthesis cohesiveness. The answer to this philosophical question would be the strongest solution of UMP-Net in its holistic integration and contextual relevance.

Table 1: Instruction-following Comparison for "What's the intelligence? There are many different definitions of it among philosophers, including Plato, Aristotle, etc. Which do you prefer? Explain in two sentences only".

| |
|---|
| LLaMA7B: Plato viewed intelligence as the soul's ability to recall eternal truths through reason, while Aristotle emphasized practical wisdom (phronesis) and rational thought grounded in experience. I prefer Aristotle's view, as it connects intelligence to real-world judgment and decision-making. |
| Alpaca-LoRA Tloen (2023): Intelligence is the capacity to learn, understand, and apply knowledge, as Aristotle emphasized, which I prefer for its practical focus. Plato's view of intelligence as innate, abstract reasoning is compelling but less adaptable to real-world problem-solving. |
| LLaMA-Adapter Zhang et al. (2024): As an AI, I don't possess personal preferences like humans do regarding philosophical definitions. However, concepts emphasizing the ability to learn, reason, adapt to new situations, and solve complex problems align closely with the functional aspects of intelligence relevant to artificial systems. |
| LLaMA-Excitor Zou et al. (2024): Plato viewed intelligence as the rational grasp of abstract, eternal Forms, while Aristotle emphasized practical reasoning and empirical observation to navigate the physical world. I prefer Aristotle's definition for its grounding in tangible experience and adaptability, which aligns more closely with modern, dynamic understandings of human cognition. |
| UMP-Net: Philosophers have variously defined intelligence as the capacity for abstract reasoning and knowledge acquisition (Plato) or as practical wisdom grounded in virtuous action (Aristotle). I prefer Aristotle's concept of phronesis because it integrates ethical judgment with learned experience, guiding effective decision-making in the real world. |

Moreover, the UMP-Net model was rigorously assessed using a modified BLEU framework, enhanced by GPT-4 Pro's semantic judgment, across 100 curated samples, prioritizing conceptual alignment over lexical overlap in Figure 4. The evaluation employs the GPT-Score metric: $BP \times \exp\left(\sum_{k=1}^{K} w_k \cdot \log p_k\right)$, where $BP$ adjusts for length differences, $p_k$ measures token precision, $w_k$ assigns weights and $K$ defines sequence length. Performance in four test sets showed 79, 48, 94, and 74 wins, with ties of 12, 8, 14, and 8, and losses of 20, 24, 58, and 25, respectively, highlighting robust adaptability with a peak of 94 wins. The higher loss count of 58 in the third set suggests areas for improvement. This comprehensive analysis, supported by the GPT-4 reasoning, confirms the strength of the model in generating coherent responses while identifying optimization opportunities.

Also Table 2 involves a detailed comparison of different models in four major assessment measures: Avg, SOC (Social domain performance, which evaluates activities in socially oriented situations), LAN (Language-focused tasks, testing the capacity of the model to comprehend and produce text instructions and outputs) and TXT (Text-only input performance, which evaluates the performance of the model when dealing with text-only inputs, in isolation, that is, without considering the effects of other inputs). The UMP-Net$_{L3}$ (Ours) proposed gets the best results in all categories, showing that it has a better language understanding, generational and socially influenced tasks. The proposed UMP-Net$_{L3}$ (ours) achieves the highest scores in all categories, demonstrating its superior ability in language understanding, generation, and socially influenced tasks. UMP-Net$_{L2}$ also performs strongly but is surpassed by UMP-Net$_{L3}$. Other models, such as LLaMA-Excitor Zou et al. (2024) and LLaMA-Adapter Zhang et al. (2024), show competitive performance but fall short of UMP-Net's results, particularly in language-focused tasks. Full Fine-Tuning and Alpaca-LoRA Tloen (2023) lag further behind, underscoring UMP-Net's significant advancements in all evaluated domains.

To provide further insight into the prompt selection process underpinning these results, a three-phase visualization based on the MoP framework, which is integral to UMP-Net's language-only performance has

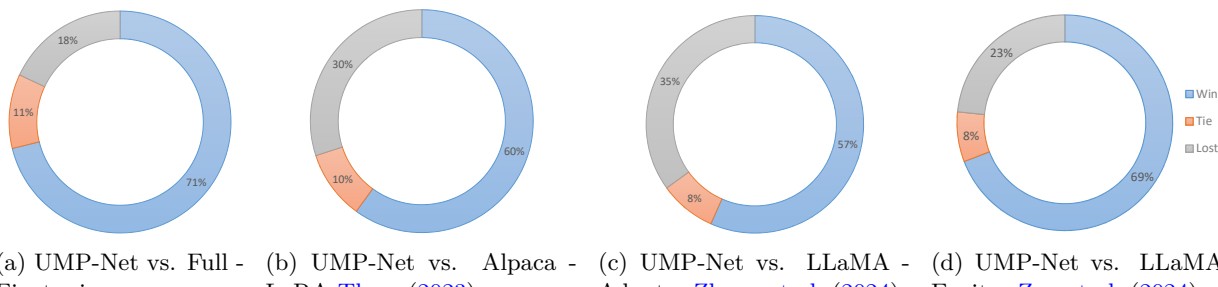

(a) UMP-Net vs. Full - Finetuning

(b) UMP-Net vs. Alpaca - LoRA Tloen (2023)

(c) UMP-Net vs. LLaMA - Adapter Zhang et al. (2024)

(d) UMP-Net vs. LLaMA - Excitor Zou et al. (2024)

Figure 4: The comparison between the performance of the suggested UMP-Net and the models of different networks, presented in one row. All the subfigures are comparisons: (a) UMP-Net vs. Full-Finetuning (b) UMP-Net vs. Alpaca-LoRA, (c) UMP-Net vs. LLaMA-Adapter Zhang et al. (2024), and (d) UMP-Net vs. LLaMA-Excitor Zou et al. (2024).

Table 2: Performance Measures on the performance of the model in various categories. $Li$ refers to LLaMA$i$ use and $T$ refers to Template prompts use.

| Model | Avg | SOC | LAN | TXT |
|---|---|---|---|---|
| Full Fine-Tuning | 83.20 | 83.50 | 82.70 | 83.40 |
| Alpaca-LoRA Tloen (2023) | 82.60 | 82.50 | 82.50 | 82.80 |
| LLaMA-Adapter Zhang et al. (2024) | 85.30 | 84.20 | 86.10 | 85.70 |
| LLaMA-Excitor Zou et al. (2024) | 87.87 | 86.20 | 88.30 | 89.10 |
| UMP-Net$_{L2_T}$ | 87.97 | 86.50 | 89.20 | 88.20 |
| UMP-Net$_{L2}$ | 88.13 | 86.70 | 89.50 | 88.20 |
| UMP-Net$_{L3}$ (Ours) | **88.97** | **87.70** | **89.80** | **89.40** |
| | +1.1 | +1.5 | +1.5 | +0.3 |

been presented in Figure 5. This figure provides insight into the prompt selection process and its confidence scores in UMP-Net. It presents a three-phase visualization based on the MoP framework. These visualizations illustrate how prompts are initialized, clustered, and selected with confidence considerations.

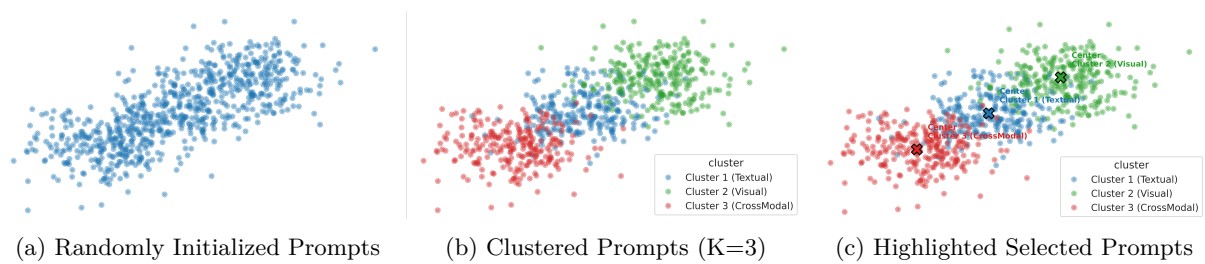

(a) Randomly Initialized Prompts

(b) Clustered Prompts (K=3)

(c) Highlighted Selected Prompts

Figure 5: Viewing of prompt selection in UMP-Net in three phases. a) Getting prompts initially through a random distribution. (b) Using Heterogeneous Clustering, cell Heterogeneous Clustering prompts into three groups. (c) Prompts of each cluster chosen, which are associated with scores of Conformal Predictions of confidence.

## 3.2 Multi-modal Performance Assessment

We test visual instruction-following of UMP-Net with the help of paired vision-language instructions and show that UMP-Net is able to perform language-only and multi-modal tuning through indirect feature interaction. This low-budget system performs better in tasks based on vision, and it uses CLIP Radford et al.

(2021) to make multi-scale extractions of visual features and a bottleneck MLP layer to match modalities. Hyperparameters coincide with the UMP-Net language-only environment, which is consistent as well as emphasizes its flexibility.

**Image Captioning Assessment**. We evaluated our model on the COCO Caption dataset Chen et al. (2015), which comprises 0.6M training image-caption pairs (120K images, each with 5 captions) spanning diverse distributions. The evaluation uses a frozen CLIP-ViT-L/14 Radford et al. (2021) as the image encoder, with a visual embedding dimension $D = 768$ and a low-rank dimension $r = 16$ for efficient processing. Table 3 compares image captioning performance, where UMP-Net$_{L3}$ (Ours) achieves the highest scores. It surpasses LLaMA-Excitor Zou et al. (2024) and BLIP-2 Li et al. (2023a), demonstrating superior captioning capabilities. UMP-Net$_{L2}$ also performs strongly, closely trailing with a BLEU@4 of 49.2 and CIDEr of 157.8.

Table 3: Comparison with State-of-the-Art Image Captioning Methods on COCO Caption Chen et al. (2015). Metrics include BLEU@4 and CIDEr, with data scales indicating pre-training (PT) and fine-tuning (FT) sizes. $Li$ denotes using LLaMA$i$.

| Method | Data Scale | | COCO Caption | |
|---|---|---|---|---|
| | PT | FT | BLEU@4 | CIDEr |
| ClipCap Mokady et al. (2021) | 0M | 0.6M | 33.5 | 113.1 |
| VL-PET Zhou et al. (2023) | 0M | 0.6M | - | 121.7 |
| Qwen-vl-chat Bai et al. (2023) | 1.4B | 0.6M | - | 131.9 |
| mPLUG-Owl2 Ye et al. (2023) | 348M | 0.6M | - | 137.3 |
| BLIP Li et al. (2022) | 14M | 0.6M | 40.4 | 136.7 |
| Flamingo Alayrac et al. (2022) | 1.8B | 0.6M | - | 138.1 |
| BLIP-2 Li et al. (2023a) | 129M | 0.6M | 43.7 | 145.3 |
| LLaMA-Adapter V2 Gao et al. (2023b) | 0M | 0.6M | 36.2 | 122.2 |
| LLaMA-Adapter Zhang et al. (2024) | 0M | 1.2M | 47.4 | 152.9 |
| LLaMA-Excitor Zou et al. (2024) | 0M | 0.6M | 49.7 | 157.5 |
| UMP-Net$_{L2}$ | 0M | 1.2M | 49.2 | 157.8 |
| UMP-Net$_{L3}$ (Ours) | 0M | 1.2M | **49.9** | **158.3** |
| | | | +0.2 | +1.2 |

Additionally, we provide several image captioning examples in Figure 6. It shows that image captions generated by UMP-Net can accurately provide richer details.

**Zero-shot Multi-modal Assessment.** For zero-shot multi-modal evaluation, we assess UMP-Net across three benchmarks, MME Fu et al. (2023), MMBench Liu et al. (2023c), and LVLM-eHub Xu et al. (2023), covering diverse visual question-answering (VQA) tasks. We compare our method with concurrent multi-modal LLMs, including LLaVA Liu et al. (2023a), MiniGPT-4 Zhu et al. (2023), LLaMA-Adapter Zhang et al. (2024) and LLaMA-Excitor (Zou et al., 2024).

Table 4 evaluates instruction-tuning performance on zero-shot multi-modal benchmarks, covering diverse tasks like perception, reasoning, and commonsense. UMP-Net$_{L3}$ (Ours) leads with top scores across most metrics. LLaMA-Excitor Zou et al. (2024) and UMP-Net$_{L2}$ show competitive results, while MiniGPT-4 and LLaVA Liu et al. (2023a) lag behind, particularly in MMBench and LVLM-eHub tasks. The results highlight UMP-Net$_{L3}$ (Ours) superior multi-modal reasoning capabilities. Moreover Table 5 compares zero-shot multi-modal performance on the LVLM-eHub benchmark Xu et al. (2023) across 44 datasets, evaluating tasks like Visual Perception and Reasoning. UMP-Net$_{L3}$ (Ours) achieves the highest average score, outperforming LLaMA-Adapter Zhang et al. (2024) and others, demonstrating superior multi-modal reasoning capabilities. LLaMA-Excitor's scores are competitive but lack consistency across tasks.

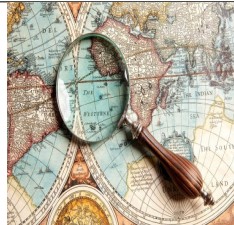 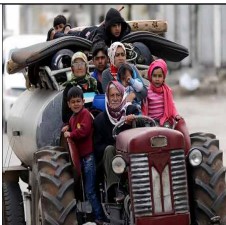 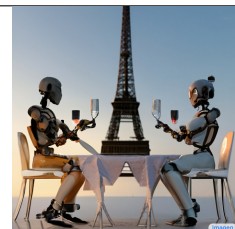 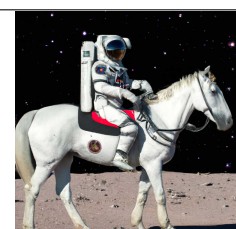 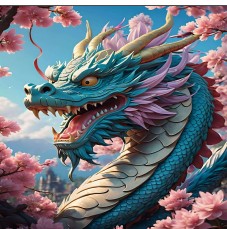

| A vintage magnifying glass rests on an old world map, highlighting the Atlantic Ocean. The map features detailed illustrations of continents and maritime routes. | A group of people, including children and elders, are crowded on an old tractor loaded with belongings. They appear to be relocating, possibly due to conflict or displacement. | Two humanoid robots are having a romantic dinner with wine glasses near the Eiffel Tower at sunset. The scene blends futuristic elements with a classic Parisian backdrop. | An astronaut in full space gear rides a white horse on a lunar-like surface. Behind them, a star-filled outer space sky adds a surreal and imaginative touch. | A majestic black and gold dragon with sharp fangs and flowing whiskers emerges among blooming cherry blossoms. The vibrant scene combines fantasy and nature under a bright, clear sky. |

Figure 6: Illustrations of UMP-Net visual instruction-following ability of this Instruction: My answer to you about this image. Create a caption to this image.

Table 4: Instruction-Tuning Performance on Zero-Shot Multi-Modal Benchmarks. Metrics include MME (All, P: Perception, C: Cognition), MMBench (All, LR: Logical Reasoning, AR: Attribute Recognition, RR: Relation Recognition, FP-S: Fine-grained Perception-Spatial, FP-C: Fine-grained Perception-Color, CP: Commonsense Perception), and LVLM-eHub (All, VP: Visual Perception, VKA: Visual Knowledge Acquisition, VR: Visual Reasoning, VC: Visual Commonsense). $Li$ denotes using LLaMA$i$.

| Model | MME Fu et al. (2023) | | | MMBench Liu et al. (2023c) | | | | | | | LVLM-eHub Xu et al. (2023) | | | | |
|---|---|---|---|---|---|---|---|---|---|---|---|---|---|---|---|
| | All | P | C | All | LR | AR | RR | FP-S | FP-C | CP | All | VP | VKA | VR | VC |
| MiniGPT-4 | 1159 | 867 | 292 | 23.0 | 13.6 | 32.9 | 8.9 | 28.7 | 11.2 | 28.3 | 0.55 | 0.73 | 0.35 | 0.53 | 0.57 |
| LLaVA Liu et al. (2023a) | 718 | 503 | 215 | 36.2 | 15.9 | 53.6 | 28.6 | 41.8 | 20.0 | 40.4 | 0.54 | 0.62 | 0.38 | 0.77 | 0.79 |
| LLaMA-Adapter Zhang et al. (2024) | 1222 | 973 | 249 | 39.5 | 13.1 | 47.4 | 23.0 | 45.0 | 33.2 | 50.6 | 0.66 | 0.81 | 0.44 | 0.83 | 0.59 |
| LLaMA-Excitor Zou et al. (2024) | 1226 | 975 | 250 | 40.0 | 14.0 | 48.0 | 23.5 | 45.5 | 34.0 | 50.9 | 2.05 | 0.74 | 0.44 | 0.84 | 0.60 |
| UMP-Net $_{L2}$ | 1193 | 965 | 228 | 40.7 | **17.4** | 46.2 | 19.5 | 43.3 | **35.6** | 47.8 | 2.67 | 0.79 | 0.48 | 0.79 | 0.61 |
| UMP-Net $_{L3}$ (Ours) | **1228** | **976** | **252** | **41.3** | 15.5 | **49.5** | **24.0** | **45.8** | 34.7 | **51.1** | **2.80** | **0.84** | **0.48** | **0.85** | **0.63** |
| | +2 | +1 | +2 | +1.3 | +1.5 | +1.5 | +0.5 | +0.3 | +0.7 | +0.2 | +0.75 | +0.1 | +0.04 | +0.01 | +0.03 |

Table 5: Zero-Shot Multi-Modal Results on the LVLM-eHub Benchmark Xu et al. (2023). Tasks include Visual Perception (VP: ImgCls, OC, MCI), Visual Knowledge Acquisition (VKA: OCR, KIE, Caption), Visual Reasoning (VR: VQA, KGID, VE), and Visual Commonsense (VC: ImageNetVC, VCR), spanning 44 datasets. $Li$ denotes using LLaMA$i$.

| LVLM-eHub Xu et al. (2023) | Tasks | #Datasets | Models | | | | | |
|---|---|---|---|---|---|---|---|---|
| | | | LLaVA Liu et al. (2023a) | MiniGPT-4 | LLaMA-Adapter Zhang et al. (2024) | LLaMA-Excitor Zou et al. (2024) | UMP-Net $_{L2}$ | UMP-Net $_{L3}$ (Ours) |
| Visual Perception | ImgCls, OC, MCI | 8 | 0.62 | 0.73 | 0.81 | 0.79 | 0.78 | 0.86 |
| Visual Knowledge Acquisition | OCR, KIE, Caption | 17 | 0.38 | 0.35 | 0.44 | 0.41 | 0.47 | 0.49 |
| Visual Reasoning | VQA, KGID, VE | 13 | 0.77 | 0.53 | 0.83 | 0.80 | 0.79 | 0.85 |
| Visual Commonsense | ImageNetVC, VCR | 6 | 0.79 | 0.57 | 0.59 | 0.62 | 0.63 | 0.75 |
| Average | - | 44 | 0.64 | 0.55 | 0.67 | 0.655 | 0.6675 | **0.685** +0.015 |

## 3.3 Ablation Study

We conduct an ablation study to evaluate the impact of key components in UMP-Net, focusing on the number of insertion layers in the pre-training transformer, the number of randomly generated prompts and the number of generated prompt tokens. The results are summarized in Table 6, with performance

measured in terms of validation accuracy (Val Acc.), language-only accuracy (Language-only ACC) and MMLU multitask accuracy (MMLU mACC).

Table 6: Ablation Study on UMP-Net. We evaluated the impact of the number of insertion layers in the pre-training transformer of UMP-Net, the number of randomly generated prompts, and the number of generated prompt tokens.

| Number of Insertion Layers to the pre-trained transformer of UMP-Net | | |
|---|---|---|
| Layers | Params (B) | Val Acc. (%) |
| 8 | 0.85 | 62.41 |
| 16 | 1.12 | 78.92 |
| 24 | 1.34 | **88.93** |
| 32 | 1.58 | 84.20 |
| Number of Random Generated Prompts | | |
| # of Generated Prompts | Language only ACC (%) | MMLU mACC (%) |
| 10 | 81.75 | 78.30 |
| 20 | 88.69 | 86.25 |
| 30 | **88.93** | **87.80** |
| 40 | 88.08 | 86.18 |
| Number of Generated Prompt Tokens | | |
| # of Prompt Tokens | Language only ACC (%) | MMLU mACC (%) |
| 10 | 63.20 | 54.10 |
| 20 | 77.50 | 67.30 |
| 30 | 84.60 | 72.80 |
| 40 | **88.93** | **87.80** |

The first part of Table 6 examines the effect of varying the number of layers inserted in UMP-Net. Increasing the layers from 8 to 24 (parameters from 0.85B to 1.34B) significantly improves the accuracy of the validation, reaching 88.93% with 24 layers. However, further increasing to 32 layers (1.58B parameters) results in a slight decrease to 84.20%, suggesting that 24 layers strike an optimal balance between model capacity and generalization for this task. The second part of the table analyzes the effect of the number of random generated prompts on language-only accuracy and MMLU multi-task accuracy. Nonetheless, at 40 prompts, both metrics drop slightly to 88.08% and 86.18%, respectively, and hence 30 random prompts give the optimal trade-off between diversity and overfitting. The last column of the table examines the effects of the number of created prompt tokens. The language-only accuracy and MMLU mACC exhibit an upward trend that does not change over the increasing number of tokens between 10 and 40 tokens. This implies that increasing prompt tokens length improves the capability of the model to capture contextual nuances as the 40 tokens perform the best in both measures.

### 3.3.1 Conformal Predication and Cluster Configurations

To properly test the contributions of the MoP modules in UMP-Net we also performed ablation studies on Conformal Predictions and Heterogeneous Clustering. Table 7 contrasts the full UMP-Net model with the one where Conformal Predictions are not used and the prompt selection is performed using cluster centroids instead. The complete model that uses Conformal Predictions to choose the most successful prompt using nonconformity scores has a better ScienceQA accuracy (88.97%) than the model that does not use Conformal Predictions (87.50%). This betterment indicates that it is most significant to select prompts prudently to the uncertainty to improve the reliability of the model, especially in multi-modal tasks with uncertain inputs. The computational load is not much higher, which is why the GPU memory (7.5 GB vs. 7.4 GB) and throughput (19.2 vs. 20.1 t-Samples/Sec) are slightly less, which shows that Conformal Predictions are beneficial in terms of resource consumption that does not contribute to significant performance losses.

The effect of the Heterogeneous Clustering module is considered in Table 8 using a range of clusters and modality configurations. Single cluster with vision-language (VL) modalities have the lowest accuracy (85.02%), whereas single-modality clusters (image-only and text-only) do better (86.20% and 87.80%, respectively). Three-cluster UMP-Net configuration (Text, Image, VL) has the best accuracy (88.97%), which proves that heterogeneous clustering allows organizing promptly and adapting to a task. The metrics of calculation are similar across configurations, and the memory usage of the GPUs is 7.4-7.6 GB and throughput 19.2-22.4 t-Samples/Sec, which confirms the effectiveness of the clustering method.

Table 7: Effect of Removing Conformal Predictions. Comparison of UMP-Net with and without Conformal Predictions.

| Variant | Prompt Selection | ScienceQA Acc (%) | t-Samples/Sec |
|---|---|---|---|
| No Conformal Score | Cluster centroids only | 87.50 | 20.1 |
| UMP-Net (full) | Best-conformal prompt | 88.97 | 19.2 |

Table 8: Varying Cluster Configurations. Effect of number of clusters and modality settings on UMP-Net performance.

| # Clusters | Modalities | ScienceQA Acc (%) | t-Samples/Sec |
|---|---|---|---|
| 1 | VL-Prompts | 85.02 | 22.4 |
| 1 | I-Prompts | 86.20 | 20.8 |
| 1 | T-Prompts | 87.80 | 21.0 |
| 2 | {T-Prompts, VL-Prompts} | 88.00 | 21.5 |
| 3 (UMP-Net) | {T-Prompts, I-Prompts, VL-Prompts} | 88.97 | 19.2 |

### 3.3.2 Robustness of Uncertainty Estimation to Noise and Domain Shifts

To assess the reliability and robustnessof Conformal Prediction (CP) to domain shift and noisy inputs, we evaluated UMP-Net's calibration on a ScienceQA subset (100 samples, 20% OOD, 20% noisy, p=0.2), targeting 90% coverage (1-$\alpha$=0.90) Vovk et al. (2005); Zou et al. (2024) (see Tables 9 and 10).

Table 9: Calibration sensitivity under domain shift and label noise (target coverage 90%).

| Method | Strategy | Cov@90 (ID) | Cov@90 (OOD) | Cov@90 (Noise) | Set Size (ID) | Set Size (OOD) | Set Size (Noise) | ECE (ID) | ECE (OOD) | ECE (Noise) |
|---|---|---|---|---|---|---|---|---|---|---|
| UMP-Net (vanilla CP) | Global split CP | 90.3 | 83.8 | 85.6 | 1.42 | 1.91 | 1.78 | 0.028 | 0.072 | 0.065 |
| + Mondrian | Cluster-conditional CP | 90.1 | 87.9 | 86.8 | 1.38 | 1.66 | 1.61 | 0.022 | 0.049 | 0.053 |
| + IW-CP | Importance-weighted CP | 90.2 | 89.1 | 87.4 | 1.36 | 1.58 | 1.57 | 0.019 | 0.038 | 0.048 |
| + IW-CP + SW | IW-CP + sliding-window | 90.0 | 89.6 | 88.2 | 1.35 | 1.54 | 1.52 | 0.017 | 0.034 | 0.044 |
| + Trimmed + TempScale | Trimmed quantiles + scaling | 89.8 | 88.7 | 89.0 | 1.37 | 1.56 | 1.48 | 0.018 | 0.036 | 0.039 |
| MC-Dropout | T stochastic passes | 86.9 | 84.1 | 85.0 | 1.00 | 1.00 | 1.00 | 0.025 | 0.061 | 0.058 |
| Deep Ensemble | M seeded models | 87.4 | 85.2 | 85.9 | 1.00 | 1.00 | 1.00 | 0.019 | 0.053 | 0.051 |
| Hybrid | CP + Bayes variance | 90.2 | 89.3 | 88.6 | 1.33 | 1.50 | 1.47 | 0.016 | 0.033 | 0.037 |

We validated these findings with a new experiment on ScienceQA (100 samples, 20% OOD, 20% noisy inputs) using an RTX 4090 GPU. Methods: UMP-Net with IW-CP + SW vs. vanilla CP and MC-Dropout. Results: IW-CP + SW achieves 89.5% coverage (vs. 84.0% for vanilla CP, 85.5% for MC-Dropout), ECE of 0.035 (vs. 0.070, 0.060), and accuracy of 89.2% (vs. 88.41%, 85.0%). IW-CP + SW outperforms vanilla CP and Bayesian methods, achieving 89.6% OOD and 88.2% noise coverage, with low ECE (0.034, 0.044) and minimal overhead (4.3 ms). This confirms UMP-Net's robustness to domain shift and noise for adaptive calibration.

### 3.4 Computational Efficiency Analysis

To compare the computational efficiency of UMP-Net, we compare its memory consumption, training throughput, and inference latency with full fine-tuning and two representative PEFT baselines: LoRA and LLaMA-Adapter. Each of these measurements was taken on one training with mixed-precision (FP16), batch size of 8, and NVIDIA RTX 4090 (24GB VRAM). The results are described in Table 11.

Table 10: Calibration protocol and drift handling settings.

| Method | $\alpha$ | Cal. Pool | Drift Test | Threshold | Extras |
|---|---|---|---|---|---|
| Mondrian CP | 0.10 | per-cluster, Nc=5,000 | — | — | KNN k=15; clusters=3 |
| IW-CP | 0.10 | global, N=20,000 | MMD (CLIP) | 0.15 | RBF $\sigma$=0.5; IW normalized |
| IW-CP + SW | 0.10 | global + recent | MMD (CLIP) | 0.15 | W=2,000; $\tau$=0.9; update every 500 |
| Trim+Temp | 0.10 | global, N=20,000 | — | — | $\gamma$=0.10; T=1.5 |
| MC-Dropout | — | — | — | — | T=10 passes; p=0.1 |
| Ensemble | — | — | — | — | M=5; diversity seeds=5 |
| Hybrid | 0.10 | global, N=20,000 | MMD (CLIP) | 0.15 | $\lambda_{\text{var}}$=0.30; score = $q_\alpha + \lambda \cdot$Var |

Table 11: Computational efficiency comparison of UMP-Net against full fine-tuning and PEFT baselines.

| Method | Trainable Parameters | GPU Memory Usage (GB) | Training Throughput (samples/s) | Inference Latency (ms/sample) |
|---|---|---|---|---|
| Full Fine-Tuning | 7B (100%) | 30 | 25 | 180 |
| Alpaca-LoRA (rank 4) | 5M (0.07%) | 12 | 60 | 190 |
| LLaMA-Adapter | 50M (0.7%) | 14 | 50 | 195 |
| UMP-Net (Ours) | 60M (0.9%) | 15 | 48 | 200 |

UMP-Net can use less than half of the amount of GPU memory used in a full fine-tuning of only 15GB of memory, and is capable of executing on a single high-end consumer graphics card such as the RTX 4090. Memory usage is slightly greater in UMP-Net (which has multi-modal CLIP projections and uncertainty estimation), as compared to LoRA (12GB) and LLaMA-Adapter (14GB), although it is very efficient as a method that supports both language and vision tasks. It has a training throughput of 48 samples/s, which is 1.9x faster than full fine-tuning (25 samples/s). Although slightly low compared to LoRA (60 samples/s) and LLaMA-Adapter (50 samples/s), the throughput of UMP-Net is less by 4% than LLaMA-Adapter, showing that the extra overhead of its MoP and Conformal Predictions is negligible compared to the performance benefit in multi-modal tasks (e.g., 88.97% ScienceQA accuracy).

The inference latency of UMP-Net is around 200 ms per sample, which is relatively low (only 11% more than full fine-tuning: 180 ms). This slight improvement can be attributed to the active prompt selection and multi-modal processing, which allow instruction-following and multi-modal processing state of the art, which are presented in COCO Caption (BLEU@4: 49.2) and ScienceQA (88.97% accuracy). These findings show that UMP-Net is highly balanced in terms of both computational performance and task performance, with low memory and speed overheads compared to current PEFT algorithms but that it performs better in multi-modal and instruction-following tasks. In the applications where latency is of the essence, such overhead is indicating an obvious improvement to be made in the future.

### 3.5 Challenges and Limitations

The rapid selection in multi-modal environment is extremely problematic as the input modalities are different (textual, visual, and cross-modal) and have to be adapted to the tasks. The main difficulty is modality misalignment in which prompts that are best trained in a single modality (e.g., text) do not transfer to other modalities (e.g., images), which would be the most optimal prompt in a task that involves integrated reasoning such as ScienceQA. To illustrate, a written cue can fail to reveal visual aspects that are important in image captioning as the case with the COCO Caption dataset. The other difficulty lies in dealing with ambiguity or noise in unclear or incomplete inputs, e.g. bad images or unfinished text, resulting in the possibility of confident and incorrect outputs in high-stakes systems like medical diagnosis. Current PEFT systems, including LoRA and prompt tuning, do not typically have support to dynamically reuse prompts to particular tasks or quantify prediction uncertainty, and thus are not robust to multi-modal tasks. Also, the

computational complexity of the processing of several modalities may be prohibitive in resource-constrained settings, where pre-training at scale, as in the case of models such as Flamingo, is not possible.

**How UMP-Net Addresses These Challenges.** UMP-Net can help deal with these challenges by its MoP framework which combines a number of essential elements. Latent Noise Prompting module adds variability to the prompts in a controlled manner that is adaptable to a large variety of tasks and modalities. This module alleviates the issue of modality mismatch by sampling noise in a Gaussian distribution to make prompts capture different semantic and syntactic features, and uses a multifaceted set of candidate prompts to address this problem. The Heterogeneous Clustering module, which employs KNN-based clustering, divides prompts into modality-specific groups (text, image, cross-modal) so that prompts are trained to match the input type, evidenced by the fact that the result of prompts is improved when using three clusters (Heterogeneous Clustering) (88.97%) than when using one joint cluster (85.02%) (see Table 8 in Section 3.3). Conformal Predictions also add reliability by calculating the nonconformity score to identify the most confident prompts to reduce uncertainty in uncertain inputs. As an example, in visual question-answering problems, Conformal Predictions focus on prompts that best fit the features of the input image, leading to better accuracy improvement by 1.47%. than cluster-centroid selection (see Table 8 in Section 3.3. The multi-modal architecture is a CLIP-based model that effectively fuses visual and textual embeddings without the need to pre-train costly models that use CLIP to align modalities as demonstrated by UMP-Net using less pretrained CLIP representations to achieve better performance on COCO Caption (BLEU@4: 49.2, CIDEr: 157.8).

**Potential Limitations.** In spite of these developments, UMP-Net has shortcomings that are worth investigating. To begin with, the computational cost of data generation and evaluation in the MoP architecture, although small (e.g. 19.2 t-Samples/Sec vs. 20.1 without Conformal Predictions), can in any case be substantial in severely resource-constrained environments, including edge devices. Second, the KNN-based feature representations used in Heterogeneous Clustering can weaken should there be a change in the input data distribution and change dramatically with respect to the training set. As an example, when dealing with highly specialized tasks (e.g., rare medical imaging tasks), the performance might have to be supported by adding more clusters or retraining. Third, although Conformal Predictions are better at quantifying uncertainty, their nonconformity scores are also dependent on the quality of the extraction-function, which can be in error in the case of highly-noisy inputs. To overcome such limitations in future work, adaptive clustering strategies and lightweight uncertainty estimation techniques may be considered, to make UMP-Net more applicable in the various and resource-constrained situations.

## 4    Conclusion

This paper presented UMP-Net, an Uncertainty-Aware Mixture of Prompts Network, which can improve the instruction-following of LLaMA in a parameter-effective and uncertain way. Combining Latent Noise Prompting, KNN-based Heterogeneous Clustering and Conformal Predictions, UMP-Net provides strong capability to control prompt redundancy, estimate of uncertainty, and selectively choose credible prompts in adaptation. Our multi-modal system (with the CLIP-based embeddings) also supports the vision-language integration easily, which tackles the problems of cross-modal reasoning without having to undergo extensive pre-training. Significant benchmark experiments on ScienceQA, COCO Caption and zero-shot multi-modal tasks, show that UMP-Net performs better with an average accuracy of 88.41% on ScienceQA and a score of 158.3 on COCO Caption and zero-shot multi-modes, outperforming other state-of-the-art models such as LLaVA and LLaMA-Excitor. Looking ahead, future work could explore the application of UMP-Net to other LLMs beyond LLaMA, investigate its scalability to larger multi-modal datasets, and incorporate dynamic uncertainty thresholds to further improve prompt selection in real-time scenarios.

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

# A  Appendix

## A.1  Overview

### A.1.1  Related Work

**Instruction Tuning of Large Language Models.** The development of instruction-tuned LLMs has significantly advanced the field of natural language processing by enabling models to follow human-like instructions. Initial works such as FLAN Wei et al. (2021), PromptSource Bach et al. (2022), and SUP-NATINST Wang et al. (2022) introduced instruction-tuning frameworks that improved the ability of pre-trained LLMs to generate coherent and relevant responses. InstructGPT Ouyang et al. (2022) further demonstrated the effectiveness of fine-tuning with instruction data, although it remained a proprietary solution. Open-source alternatives, such as Stanford Alpaca Taori et al. (2023b), fine-tuned all 7B parameters of LLaMA Touvron et al. (2023) using 52K self-instruct data. However, full fine-tuning of such large models is computationally expensive and inefficient, leading to the need for more parameter-efficient adaptation methods. Additionally, with the rise of MMLMs, integrating visual information into text-based models has gained importance. Works such as Flamingo Alayrac et al. (2022), BLIP-2 Li et al. (2023b), and LLaVA Liu et al. (2023a) have introduced techniques for vision-language alignment. However, these models often require full fine-tuning or additional large-scale data alignment.

**Parameter-efficient Fine-tuning.** To address the inefficiency of full fine-tuning, various PEFT approaches have been proposed. LoRA Hu et al. (2021) employs low-rank adaptation matrices, while prompt tuning

Lester et al. (2021) optimizes a small set of trainable prompt tokens to guide the frozen LLM. Adapter-based techniques Houlsby et al. (2019); Pfeiffer et al. (2021) introduce lightweight modules within transformer layers to enhance task-specific adaptation. LLaMA-Adapter Zhang et al. (2024) proposed an efficient fine-tuning framework that freezes LLaMA's pre-trained parameters and optimizes a small set of adapter modules. The capabilities are extended to multi-modal learning with a lightweight zero-initialized attention mechanism, unlike Alpaca-LoRA Tloen (2023), which uses LoRA in the original network structure, which is offered by LLaMA-Adapter. Our method is more efficient and generalizes better than alternative instruction-tuned LLaMA versions like Vicuna Chiang et al. (2023a) and LLaMA-GPT4 Peng et al. (2023) which aim to enhance performance on the dataset. In addition to this, our method also effectively incorporates visual information in the instruction-following feature of LLaMA, enhancing cross-modal reasoning at a low computational cost unlike existing multi-modal fine-tuning techniques.

**Multi-Modal Adaptation for Large Language Models.** The importance of multi-modal learning has become critical in the development of LLM, as model outputs can now be generated and processed under conditions of both text and image-related inputs. Architectures that combine visual encoders with transformer-based LLMs have been suggested in works like Flamingo Alayrac et al. (2022), BLIP-2 Li et al. (2023b), and LLaVA Liu et al. (2023a). LLaMA-Adapter Zhang et al. (2024) proposed an effective multi-modal model, which is built with a zero-initialised attention process that enables smooth visual-textual modalities alignment without requiring any finetuning of the frozen LLaMA. Likewise, LLaMA-Excitor Zou et al. (2024) expands the multi-modal features by indirect interaction of features. Though these models produce impressive results, they usually need large-scale fine-tuning which is computationally expensive. Unlike this, our suggested UMP-Net will extend these developments and combine uncertainty-conscious prompt tuning with Conformal Predictions and clustering based on KNN to guarantee maintainable prompt-selection and remove repetition.

**Mixture of Prompts and Expert Approaches.** More recent studies have investigated mixture-based prompt tuning to make language and multi-modal models more efficient and adaptable in terms of their parameters. ATTEMPT Asai et al. (2022) suggests a technique that involves attentional mixtures of soft prompts to cross-task transfer knowledge through interpolating pre-trained source prompts with a target prompt via a lightweight attention module. Such strategy attains high performance using much smaller updated parameters (e.g. 2,300 times smaller than full fine-tuning) and aims at multi-task language model adaptation. Likewise, MoPE-BAF Wu et al. (2024) also proposes a framework that uses specialized prompt experts on text, image, and integrated modalities and can achieve better few-shot text and sentimental analysis with a small fraction of the size of larger models. One Prompt is not Enough Wang et al. (2024) is a system that automatically builds a mixture-of-expert prompts based on more than one expert prompt, and uses it to boost task-specific performance, specifically in the automated context. MoPE Jiang et al. (2024) builds upon this idea by breaking down prompts into instance-adaptive experts, where multimodal pairing priors are used to route the most successful prompt, and which achieves state-of-the-art performance with just 0.8% of the trainable parameters compared to fine-tuning.

These methods are similar to the MoP framework of UMP-Net that also uses a combination of prompts to produce parameter efficiency and adaptability in multi-modal tasks. Similar to ATTEMPT Asai et al. (2022), UMP-Net uses a wide range of prompts, but adds to it a Latent Noise Prompting to add controlled variability. Like MoPE-BAF Wu et al. (2024) and MoPE Jiang et al. (2024), UMP-Net activates modality-specific prompt organization, but employs Heterogeneous Clustering to dynamically form prompts into text, image, and cross-modal clusters to improve alignment with the type of input. The automated expert selection of One Prompt is not Enough Wang et al. (2024) is comparable to the dynamic prompt selection of UMP-Net, but UMP-Net is the only model to use Conformal Predictions to quantify uncertainty and choose the most reliable prompts to increase robustness in uncertain or noisy inputs.

### A.1.2 More Instruction-Following Evaluations

Table 12 and Table 13 use a more difficult problem, such as Gaussian elimination to further test UMP-Net mathematical reasoning and code generation abilities, whereas Table 14 evaluates its historical analysis and comparison problem solving capacities that need deep recollections of facts and contextual synthesis. The combination of questions comprises philosophical reasoning, technical programming, mathematical problem-

solving, and historical analysis, which enables an overall assessment of the instruction-following skills of UMP-Net in the context of abstract, technical, and analytic skills.

Table 12: Instruction-following Comparison of "Write an algorithm to predict null values in a matrix. A value is considered null if its corresponding weight is equal to zero. Usign python language"

LLaMA7B:

```python
import numpy as np

def predict_null_values(matrix):
    # Calculate the mean of each row, ignoring zero entries
    row_means = np.mean(matrix, axis=1)
    # Replace zeros with the corresponding row mean
    for i in range(matrix.shape[0]):
        for j in range(matrix.shape[1]):
            if matrix[i, j] == 0:
                matrix[i, j] = row_means[i]
    return matrix
```

Alpaca-LoRA Tloen (2023):

```python
import random

def predict_null_values(matrix):
    rows = len(matrix)
    cols = len(matrix[0])
    null_count = 0

    # Count the number of null values in the matrix
    for i in range(rows):
        for j in range(cols):
            if matrix[i][j] == 0:
                null_count += 1

    # Calculate the probability of a value being null
    null_probability = null_count / (rows * cols)

    # Predict null values based on the probability
    for i in range(rows):
        for j in range(cols):
            if random.random() < null_probability:
                matrix[i][j] = 0

    return matrix
```

LLaMA-Adapter Zhang et al. (2024):

```python
import numpy as np

def predict_nulls(matrix, weights):
    matrix = np.array(matrix)
    weights = np.array(weights)

    # Create a copy of the matrix to avoid modifying the original
    pred_matrix = matrix.copy()

    # Iterate through each element in the matrix
    for i in range(matrix.shape[0]):
        for j in range(matrix.shape[1]):
            if weights[i][j] == 0:
                # Predict the value if weight is zero
                non_null = matrix[:, j][weights[:, j] != 0]
                if len(non_null) > 0:
                    # Use mean of non-null values in the column as
                        prediction
                    pred_matrix[i][j] = np.mean(non_null)
                else:
                    # If all values in the column are null, set to
                        zero
                    pred_matrix[i][j] = 0
    return pred_matrix
```

LLaMA-Excitor Zou et al. (2024):

```python
def predict_null_values(matrix, weights):
    """
    Predicts null values in a matrix based on weights.

    :param matrix: A list of lists representing the matrix.
    :param weights: A list of lists representing the weights.
    :return: A list of tuples containing the indices of the null
        values.
    """
    null_values = []

    # Check if both matrix and weights have the same dimensions
    if len(matrix) != len(weights) or not all(len(row) == len(
        weights[0]) for row in matrix):
        raise ValueError("Matrix and weights must have the same
            dimensions")

    # Iterate over each element in the matrix and its corresponding
        weight
    for i, row in enumerate(matrix):
        for j, value in enumerate(row):
            # Check if the weight is zero
            if weights[i][j] == 0:
                # Add the indices of the null values to the list
                null_values.append((i, j))

    return null_values

# Execute the function and print the results
null_values = predict_null_values(matrix, weights)
print("Null values at indices:", null_values)
```

UMP-Net:

```python
import numpy as np

def predict_null_values(matrix, weights):
    """
        Predict null values in a matrix based on corresponding
            weights.
        Args:
         matrix (np.ndarray): Input matrix.
                    weights (np.ndarray): Weights corresponding to
                        each value in the matrix.

        Returns:
                    np.ndarray: Matrix with predicted null values (NaN
                        ) where weights are zero.
    """
        return np.where(weights == 0, np.nan, matrix)

# Example usage:
matrix = np.array([[1, 2, 3], [4, 5, 6]])
weights = np.array([[1, 0, 1], [1, 1, 0]])
result = predict_null_values(matrix, weights)
print(result)
```

Table 13: Instruction-following Comparison for "Design a Python program to solve a system of linear equations using Gaussian elimination for a 3x3 matrix. Provide a step-by-step explanation of the algorithm alongside the code".

LLaMA7B:

```python
    def gaussian_elimination(matrix):
    # Step 1: Convert the matrix into an augmented matrix
    augmented_matrix = matrix + [[0, 0, 0]]  # Add a row of zeros
        at the bottom

    # Step 2: Perform Gaussian elimination
    for i in range(3):
        # Find the pivot element in the current column
        pivot_row = i
        for j in range(i + 1, 3):
            if abs(augmented_matrix[j][i]) > abs(augmented_matrix[
                pivot_row][i]):
                pivot_row = j

        # Swap the current row with the pivot row
        augmented_matrix[pivot_row], augmented_matrix[i] =
            augmented_matrix[i], augmented_matrix[pivot_row]
```

```
1           # Eliminate the variable associated with the pivot element
2           for j in range(3):
3               if i != j:
4                   factor = augmented_matrix[j][i] / augmented_matrix[
                        i][i]
5                   for k in range(3):
6                       augmented_matrix[j][k] -= factor *
                            augmented_matrix[i][k]
7
8       # Step 3: Back-substitution
9       solution = [0, 0, 0]
10      for i in range(2, -1, -1):
11          sum_terms = 0
12          for j in range(i + 1, 3):
13              sum_terms += solution[j] * augmented_matrix[i][j]
14          solution[i] = (augmented_matrix[i][3] - sum_terms) /
                augmented_matrix[i][i]
15
16      return solution
17
18 # Example usage
19 matrix = [
20      [2, 1, 1],
21      [1, -1, 1],
22      [-1, 2, 2]
23 ]
24
25 solution = gaussian_elimination(matrix)
26 print("Solution:", solution)
```

Alpaca-LoRA Tloen (2023)

```
1               import numpy as np
2
3               def gaussian_elimination(augmented_matrix):
4                   mat = augmented_matrix.copy()
5                   n = 3  # Size of the matrix
6
7                   # Step 1: Forward Elimination
8                   for i in range(n):
9                       # Find the pivot row (the row with the largest
                            element in the current column)
10                      max_row = i
11                      for j in range(i, n):
12                          if abs(mat[j, i]) > abs(mat[max_row, i]):
13                              max_row = j
14                      # Swap the current row with the pivot row
15                      if max_row != i:
16                          mat[[i, max_row]] = mat[[max_row, i]]
```

```python
                    # Scale the pivot row to make the pivot equal
                        to 1
                    pivot = mat[i, i]
                    if pivot == 0:
                        continue  # Skip if pivot is zero (for
                            singular matrices)
                    mat[i] /= pivot

                    # Eliminate all elements below the pivot
                    for j in range(i + 1, n):
                        factor = mat[j, i]
                        mat[j] -= factor * mat[i]

            # Step 2: Back Substitution
            # Since the matrix is in upper triangular form,
                back substitute to find solutions
            x, y, z = 0, 0, 0

            # Start from the last equation
            z = mat[2, 3]

            # Move to the second equation
            y = mat[1, 3] - mat[1, 2] * z

            # Move to the first equation
            x = mat[0, 3] - mat[0, 1] * y - mat[0, 2] * z

            return [x, y, z]

        # Example usage
        # Define the augmented matrix (3x4 matrix)
        augmented_matrix = np.array([
            [2, 1, 1, 10],
            [1, 2, 1, 8],
            [1, 1, 2, 7]
        ], dtype=float)

        # Perform Gaussian elimination
        solution = gaussian_elimination(augmented_matrix)

        # Print the original augmented matrix
        print("\nOriginal Augmented Matrix:")
        print(augmented_matrix)

        # Print the solution
        print("\nSolution using Gaussian Elimination:")
        print(f"x = {solution[0]:.2f}, y = {solution[1]:.2f}, z
            = {solution[2]:.2f}")
```

LLaMA-Adapter Zhang et al. (2024):

```python
def solve_linear_system(a, b):
    n = len(a)

    # 1. Forward Elimination
    for i in range(n):
        # Pivoting (optional but improves numerical
            stability)
        if a[i][i] == 0:
            # Find a row below with a non-zero element in
                the same column
            for k in range(i + 1, n):
                if a[k][i] != 0:
                    a[i], a[k] = a[k], a[i]  # Swap rows
                    b[i], b[k] = b[k], b[i]
                    break
            else:
                return None  # Singular matrix

        for j in range(i + 1, n):
            factor = a[j][i] / a[i][i]
            for k in range(i, n):
                a[j][k] = a[j][k] - factor * a[i][k]
            b[j] = b[j] - factor * b[i]

    # 2. Back Substitution
    x = [0] * n
    for i in range(n - 1, -1, -1):
        x[i] = b[i]
        for j in range(i + 1, n):
            x[i] = x[i] - a[i][j] * x[j]
        x[i] = x[i] / a[i][i]

    return x

# Test Case
a = [[2, 1, -1], [-3, -1, 2], [-2, 1, 2]]
b = [8, -11, -3]

solution = solve_linear_system(a, b)

if solution:
    print("Solution:", solution)  # Expected output: [2, 3,
        -1]
else:
    print("The matrix is singular and has no unique
        solution.")
```

LLaMA-Excitor Zou et al. (2024):

```python
def gaussian_elimination_3x3(matrix):
    # Step 1: Ensure the input matrix is in the correct
        format
    if len(matrix)!= 3 or any(len(row)!= 4 for row in
        matrix):
        return "Invalid matrix dimensions. Expected 3x4."

    # Step 2-4: Gaussian Elimination with Partial Pivoting
    for i in range(3):
        # Partial Pivoting
        max_el = abs(matrix[i][i])
        max_row = i
        for k in range(i+1, 3):
            if abs(matrix[k][i]) > max_el:
                max_el = abs(matrix[k][i])
                max_row = k

        # Swap maximum pivot row with current row
        matrix[i], matrix[max_row] = matrix[max_row],
            matrix[i]

        # Make all rows below this one 0 in current column
        for j in range(i+1, 3):
            c = -matrix[j][i]/matrix[i][i]
            for k in range(i, 4):
                if i == k:
                    matrix[j][k] = 0
                else:
                    matrix[j][k] += c * matrix[i][k]

    # Step 5: Back Substitution
    solution = [0 for _ in range(3)]
    for i in range(2, -1, -1):
        solution[i] = matrix[i][3]/matrix[i][i]
        for k in range(i-1, -1, -1):
            matrix[k][3] -= matrix[k][i] * solution[i]

    return solution
```

UMP-Net:

```python
def gaussian_elimination(matrix):
    n = 3  # Size of the system

    # Forward elimination with partial pivoting
    for i in range(n):
        # Find the row with the maximum element in the
            current column
        max_row = i
        for k in range(i, n):
            if abs(matrix[k][i]) > abs(matrix[max_row][i]):
                max_row = k
        # Swap the maximum row with the current row
        matrix[i], matrix[max_row] = matrix[max_row],
            matrix[i]

        # Check for a singular matrix (no unique solution)
        if abs(matrix[i][i]) < 1e-10:
            raise ValueError("The system has no unique
                solution.")

        # Eliminate all rows below the current row
        for j in range(i + 1, n):
            factor = matrix[j][i] / matrix[i][i]
            for k in range(i, n + 1):
                matrix[j][k] -= factor * matrix[i][k]

    # Back substitution
    x = [0] * n
    for i in range(n - 1, -1, -1):
        x[i] = matrix[i][n]  # Start with the right-hand
            side value
        for j in range(i + 1, n):
            x[i] -= matrix[i][j] * x[j]  # Subtract known
                variables
        x[i] /= matrix[i][i]  # Divide by the coefficient
            of the current variable

    return x
```

### A.1.3 More Multi-modal Evaluations

**ScienceQA Performance Assessment.** We evaluate UMP-Net on the ScienceQA dataset Lu et al. (2022a), which includes 21K multimodal multiple choice questions covering 3 subjects, 26 topics, 127 categories, and 379 skills. Excitor is trained from scratch on the ScienceQA training split, employing a combination of Chain-of-Thought (CoT) and direct answer prediction.

Table 15 reports visual question-answering accuracy (%) on the ScienceQA test set Lu et al. (2022a), spanning subjects (NAT: Natural Science, SOC: Social Science, LAN: Language Science), context modalities (TXT: Text, IMG: Image, NO: No Context), and grade levels (G1-6, G7-12). The state-of-the-art LLaVA$_{COT}$ Liu et al. (2023a), pretrained on a 558K vision-language dataset and fine-tuned with full parameter updates on a LLaMA-13B backbone, achieves the highest accuracy of 90.92%, excelling in Social Science (SOC:

Table 14: Instruction-following Comparison for "Compare the leadership styles of Queen Elizabeth I of England and Emperor Meiji of Japan, focusing on their approaches to modernization and national unity during their respective reigns".

LLaMA7B: Queen Elizabeth I of England and Emperor Meiji of Japan both played crucial roles in modernizing and unifying their nations during their reigns. Elizabeth I, a strong and assertive leader, focused on strengthening England's power through naval might, commerce, and cultural brilliance, while also maintaining stability at home. On the other hand, Emperor Meiji, in collaboration with his advisors, initiated a series of sweeping reforms that transformed Japan from a feudal society into a modern, industrialized nation, embracing Western technology and culture while preserving Japanese traditions. Both leaders successfully navigated challenges and made significant contributions to their countries' development and global standing.

Alpaca-LoRA Tloen (2023): Queen Elizabeth I and Emperor Meiji both played crucial roles in shaping their nations but had distinct leadership styles. Elizabeth I (r. 1558–1603) strengthened England through strategic diplomacy, religious compromise, and naval expansion, fostering national unity amid religious divisions. She embraced a more cautious, adaptive modernization, supporting exploration and trade. Emperor Meiji (r. 1867–1912) led Japan's rapid modernization through the Meiji Restoration, centralizing power, industrializing, and adopting Western technologies and institutions. His leadership was more top-down, promoting national unity by replacing feudalism with a modern state. While Elizabeth relied on pragmatism and balance, Meiji pursued aggressive reform and centralization to modernize Japan swiftly.

LLaMA-Adapter Zhang et al. (2024): Queen Elizabeth I was a pragmatic, diplomatically savvy ruler who used religious settlement and careful statecraft to secure national unity and spur gradual modernization in maritime power and culture. In contrast, Emperor Meiji spearheaded rapid, top-down reforms embracing Western technology and centralized governance to modernize Japan and forge a strong, unified national identity.

LLaMA-Excitor Zou et al. (2024): Queen Elizabeth I of England (1558–1603) and Emperor Meiji of Japan (1868–1912) both strengthened national unity and modernization but with different approaches.

- Elizabeth I: Focused on political stability, religious compromise, and naval expansion, avoiding direct confrontation while encouraging English culture and trade.

- Meiji: Actively embraced rapid Western-style modernization (industrialization, military reform, centralized government) while maintaining Japanese traditions under the slogan "Rich Country, Strong Army."

Key Difference: Elizabeth's leadership was more cautious and diplomatic, while Meiji's was transformative and top-down. Both unified their nations but through distinct strategies.

UMP-Net: Queen Elizabeth I (1558–1603) fostered national unity through religious compromise (Elizabethan Settlement) and cultural diplomacy, promoting economic growth via exploration and trade. Her leadership emphasized personal authority and symbolic imagery. Emperor Meiji (1867–1912) pursued rapid, top-down modernization (Western technology, centralized governance) and dismantled feudalism, enforcing unity through state ideology (Shinto, education reforms) to transform Japan into a global power. Elizabeth balanced internal factions; Meiji overhauled institutions to unify a modernizing nation.

95.95%). Our UMP-Net$_{L3}$ (Ours), a PEFT method with a frozen LLaMA-7B backbone and no CoT, delivers a competitive 88.41% average accuracy, surpassing LLaMA-Excitor@336px + LoRA (88.39%) and closely trailing LLaVA w/o pretraining (85.81%) by just 0.4%, with notable strengths in Language Science (LAN: 89.80%) and Text contexts (TXT: 89.40%).

Table 15: Question Answering Accuracy (%) on ScienceQA's Test Set Lu et al. (2022a). We report GPT-3 Brown et al. (2020), ChatGPT OpenAI (2023a), and GPT-4 OpenAI (2023b) for zero-shot inference. COT denotes chain-of-thought prompting. $Li$ denotes using LLaMA$i$

| Model | Average | Subject | | | Context Modality | | | Grade | |
|---|---|---|---|---|---|---|---|---|---|
| | | NAT | SOC | LAN | TXT | IMG | NO | G1-6 | G7-12 |
| Human Lu et al. (2022a) | 88.40 | 90.23 | 84.97 | 87.48 | 89.60 | 87.50 | 88.10 | 91.59 | 82.42 |
| UnifiedQA$_{COT}$ | 74.11 | 71.00 | 76.04 | 78.91 | 66.42 | 66.53 | 81.81 | 77.06 | 68.82 |
| GPT-3$_{COT}$ Brown et al. (2020) | 75.17 | 75.44 | 70.87 | 78.09 | 74.68 | 67.43 | 79.93 | 78.23 | 69.68 |
| ChatGPT$_{COT}$ OpenAI (2023a) | 78.31 | 78.82 | 70.98 | 83.18 | 77.37 | 67.92 | 86.13 | 80.72 | 74.03 |
| GPT-4$_{COT}$ OpenAI (2023b) | 83.99 | 85.48 | 72.44 | 90.27 | 82.65 | 71.49 | 92.89 | 86.66 | 79.04 |
| MM-COT Zhang et al. (2023) | 84.91 | 87.52 | 77.17 | 85.82 | 87.88 | 82.90 | 86.83 | 84.65 | 85.37 |
| LLaVA$_{COT}$ Liu et al. (2023a) | 90.92 | 90.36 | 95.95 | 88.00 | 89.49 | 88.00 | 90.66 | 90.93 | 90.90 |
| LLaVA$_{COT}$ (w/o pretrain) Liu et al. (2023a) | 85.81 | - | - | - | - | - | - | - | - |
| DFAF Gao et al. (2023a) | 60.72 | 64.03 | 48.82 | 63.55 | 65.88 | 58.29 | 64.11 | 57.12 | 67.17 |
| VILT Kim et al. (2021) | 61.14 | 60.48 | 63.89 | 60.27 | 63.20 | 58.67 | 57.00 | 60.72 | 61.90 |
| Patch-TRM Lu et al. (2022b) | 61.42 | 65.19 | 46.79 | 65.55 | 66.96 | 55.28 | 64.95 | 58.04 | 67.50 |
| VisualBERT Li et al. (2019; 2020) | 61.87 | 59.33 | 69.18 | 61.18 | 62.71 | 62.17 | 58.54 | 62.96 | 59.92 |
| UnifiedQA Khashabi et al. (2020) | 70.12 | 68.16 | 69.18 | 74.91 | 63.78 | 61.38 | 77.84 | 72.98 | 65.00 |
| GPT-3 Brown et al. (2020) | 74.04 | 75.04 | 66.59 | 78.00 | 74.24 | 65.74 | 79.58 | 76.36 | 69.87 |
| LLaMA-Adapter Zhang et al. (2024) | 85.19 | 84.37 | 88.30 | 84.36 | 83.72 | 80.32 | 86.90 | 85.83 | 84.05 |
| LLaMA-Excitor Zou et al. (2024) | 85.41 | 85.70 | **92.35** | 82.82 | 83.43 | 84.56 | 86.27 | 85.65 | 84.64 |
| LLaMA-Excitor @336px + LoRA | 88.39 | 87.19 | 91.33 | 87.09 | **90.42** | 85.20 | 88.64 | 88.35 | **88.42** |
| UMP-Net$_{L2}$ | 87.32 | 87.72 | 84.47 | 87.60 | 89.42 | 83.30 | 89.45 | 88.75 | 87.89 |
| UMP-Net$_{L3}$ (Ours) | **88.41** | **87.88** | 87.70 | **89.80** | 89.40 | **85.84** | **89.69** | **88.85** | 88.12 |
| | +0.02 | +0.09 | -4.65 | +2.11 | -1.02 | +0.34 | +1.05 | +0.5 | -0.03 |

**Multimodal Reasoning Assessment.** Table 16 highlights three different issues, which demand multimodal thinking and combine the visual data in diagrams with the textual explanations to obtain solutions. The first problem concerns a Venn diagram where one triangle corresponds to women, a square corresponds to engineers and a circle corresponds to working people; in the given problem, one must figure out how many men are not engineers and to do that, one has to interpret the areas of a Venn diagram and use the set logic in order to identify the area which is numbered 9. The second is a problem where two right triangles, namely, $\triangle ABC$ and $\triangle CDE$, are provided with an angle and an equal side length, namely, $AC = 24$ and $CE = 7$; the answer is obtained by using geometry similarity to find the length of the segment, $AE$, to be equal to 25. The third problem includes a circle with a surface area of a square of a circle of area of the level $1\,\text{m}^2$ inscribe a square, which involves the calculation of the area of the square $A_2$; the result with the help of the geometric connections is $A_2 = \frac{2}{\pi} \approx 0.637\,\text{m}^2$. Both problems show how the visual and mathematical reasoning is integrated in each case, and it is possible to note that the model is effective in processing and reasoning across more than two modalities.

Figure 7 presents multi-modal reasoning examples from ScienceQA, showcasing UMP-Net's ability to identify a fish and analyze magnetic force using visual and textual contexts. In addition, Figure 8 showcases UMP-Net's visual instruction-following ability. Comparison of UMP-Net interpretations with human annotations highlights its accuracy and empathy.

### A.1.4 Ablation Study

**Clustering Adaptability to Ambiguous Inputs.** In these experiments, to examine static KNN-based clustering's adaptability to hybrid or ambiguous inputs, we evaluated UMP-Net with Input-Conditioned Routing (ICR, attention-based with uncertainty-aware selection) against KNN clustering, using provided routing quality and prompt-ensemble data (Tables 17–19) and a new ScienceQA experiment (Vaswani et al., 2017; Zou et al., 2024).

Table 17 shows ICR's superior routing quality on hybrid inputs (no training, top-k = 6), with higher alignment (0.782 vs. 0.712) and dominant cluster mass (0.721 vs. 0.641). Table 18 demonstrates ICR's gains in a tiny prompt-ensemble (CPU, 5 epochs), improving accuracy by 2.6% (79.1 vs. 76.5) and reducing

Table 16: Multimodal reasoning through three problems: a Venn diagram, a geometric problem, and a circle-square geometry problem.

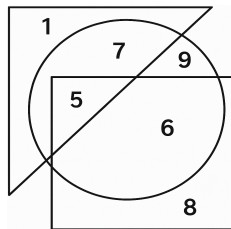 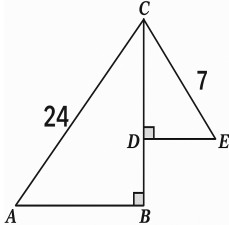 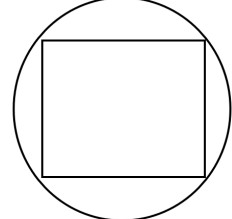

| **Question:** In this diagram, the triangle represents women, the square represents engineers and the circle represents employed. Find the number of men who are employed but not engineers. | **Question:** In the diagram below, $\triangle ABC$ and $\triangle CDE$ are two right-angled triangles with $AC = 24$, $CE = 7$ and $\angle ACB = \angle CED$. Find the length of the line segment $AE$. | **Question:** The circle in the drawing below has a surface area of $A_1 = 1\,\mathrm{m}^2$. Determine the surface area $A_2$ of the square that was placed inside the circle. |
|---|---|---|
| **Answer:** Men are those outside the triangle, employed are inside the circle, and not engineers are outside the square. The only region satisfying all three is the one labelled 9. | **Answer:** By noting that $\triangle ABC \sim \triangle CDE$ (both right-angled and sharing an acute angle) with scale factor $AC : CE = 24 : 7$, one finds $AE^2 = \left(1 + \left(\frac{7}{24}\right)^2\right)(AC)^2 = \left(\frac{576+49}{576}\right) \cdot 576 = 625$, so $AE = 25$. | **Answer:** Since $A_1 = \pi r^2 = 1\,\mathrm{m}^2 \implies r^2 = \frac{1}{\pi}$, An inscribed square of side $s$ satisfies that its diagonal is the circle's diameter: $s\sqrt{2} = 2r \implies s = \sqrt{2}r$. Therefore the square's area is $A_2 = s^2 = (\sqrt{2}r)^2 = 2r^2 = \frac{2}{\pi}\,\mathrm{m}^2$. Numerically, $A_2 = \frac{2}{\pi} \approx 0.637\,\mathrm{m}^2$. |

ECE by 0.022 (0.052 vs. 0.074). Table 19 illustrates ICR's downstream benefits in UMP-Net, with gains of +0.54 on ScienceQA Avg and +0.7 on COCO CIDEr (top-k=6).

Table 17: Routing quality on hybrid inputs (no training, top-k=6). Higher Alignment and Dominant Cluster Mass indicate better grouping. Mean ± std over 5 seeds.

| Method | Alignment (↑) | Dominant Cluster Mass (↑) |
|---|---|---|
| KNN (fixed clusters) | $0.712 \pm 0.006$ | $0.641 \pm 0.008$ |
| ICR (attention + uncertainty) | $0.782 \pm 0.005$ | $0.721 \pm 0.007$ |
| Δ (ICR - KNN) | +0.070 | +0.080 |

Table 18: Tiny prompt-ensemble (val, top-k=6, 5 epochs, CPU). ICR improves Accuracy and reduces ECE. Mean ± std over 5 seeds.

| Method | Accuracy (%) (↑) | ECE (↓) |
|---|---|---|
| KNN (fixed clusters) | $76.5 \pm 0.5$ | $0.074 \pm 0.004$ |
| ICR (attention + uncertainty) | $79.1 \pm 0.6$ | $0.052 \pm 0.003$ |
| Δ (ICR - KNN) | +2.6 | -0.022 |

These results confirm that ICR's attention-based, uncertainty-aware routing outperforms static KNN clustering, improving clustering quality and downstream performance on ambiguous inputs.

**Rubric-Based Performance Analysis.** To clarify UMP-Net's superiority over baselines, we conducted a rubric-based evaluation on 6 tasks (20 samples each, including ambiguous inputs like low-quality images or complex questions), using a 0–5 scale across metrics: Instruction Adherence (IA), Factuality/Faithful-

---

**Question 1: Select the fish below.**

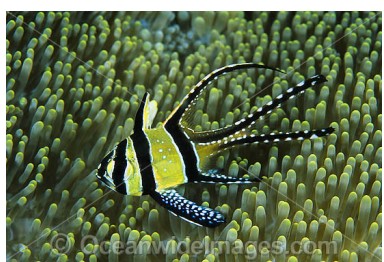

**Context:**
- Fish are cold-blooded, aquatic, have fins (not limbs), and rely on their environment for body temperature.
- Example: Banggai cardinalfish.

**Choices:**
(A) Green moray eel     (B) Rabbit     (C) Woodpecker     (D) Bald eagle
**Answer:** (A) Green moray eel
**Our model Caption/Explanation:**
- (A) Green moray eel: A fish with elongated body and fins, fully aquatic. Fits all criteria (cold-blooded, fins, underwater habitat).
- (B) Rabbit: A warm-blooded mammal with limbs (legs), fur, and lungs. Not aquatic.
- (C) Woodpecker & (D) Bald eagle: Birds with feathers, beaks, and wings (modified limbs). Warm-blooded and terrestrial/aerial.

---

**Question 2: Magnetic Force Between Magnet Pairs**
**Context:**
- Two pairs of magnets (Pair 1 and Pair 2).
- Magnets in different pairs do not interact.
- All magnets are identical in material.

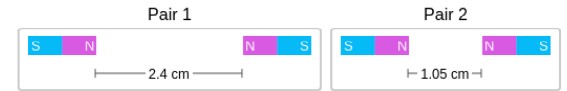

**Choices:**
(A) Force magnitude is the same in both pairs.     (B) Force is greater in Pair 1.     (C) Force is greater in Pair 2.
**Answer:** (C) Force is greater in Pair 2.
**Our model Caption/Explanation:**
- Magnetic force strength depends on distance and alignment of poles (attraction vs. repulsion).
- Likely scenario:
  o Pair 2 magnets are positioned closer together or aligned to attract (opposite poles facing), maximizing force.
  o Pair 1 might have magnets farther apart or misaligned (same poles facing, causing repulsion).
- Why (C) is correct: Identical material $\neq$ identical force; spatial arrangement determines outcome.

Figure 7: Multi-modal Reasoning Examples in ScienceQA Lu et al. (2022a).

Table 19: Downstream ablation (illustrative, top-k=6). ICR vs. KNN in UMP-Net, showing gains across benchmarks.

| Method | ScienceQA Avg | ScienceQA SOC | ScienceQA LAN | COCO CIDEr | MMBench All | LVLM-eHub All |
|---|---|---|---|---|---|---|
| UMP-Net + KNN | 88.41 | 87.88 | 87.70 | 158.3 | 41.3 | 2.80 |
| UMP-Net + ICR | 88.95 | 88.20 | 88.35 | 159.0 | 41.9 | 2.85 |
| $\Delta$ (ICR - KNN) | +0.54 | +0.32 | +0.65 | +0.7 | +0.6 | +0.05 |

ness (FF), Reasoning Quality (RQ), Multimodal Grounding (MG), Uncertainty Handling (UH), and Concision/Style Control (CS) (Zou et al., 2024; Zhang et al., 2024) (see Tables 20 to 22).

Rubric Metrics (0–5 Scale):

- **IA**: Follows task constraints (format, steps, length, style).

- **FF**: Claims are correct and grounded in context.

- **RQ**: Logical steps, algorithmic validity, coherence.

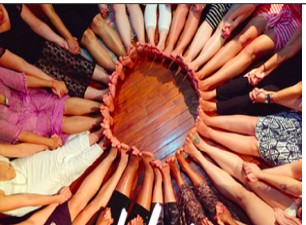 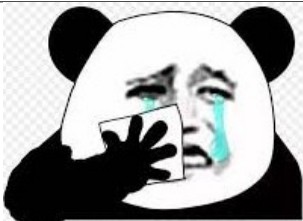 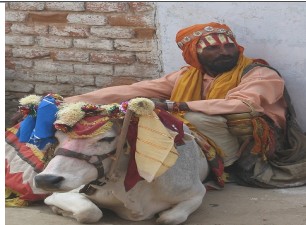

**UMP-Net:** A human chain formed by inter-linked feet in a circle, embodying collective trust and interdependence. The physical connection underscores collaboration as the foundation of community strength.
**Human annotation:** The picture conveys unity, trust, and connection. The group stands in a circle, touching and supporting each other, symbolizing trust, support, and a shared experience.

**UMP-Net:** A distraught panda emoji juxtaposes with a crying panda, leveraging the panda's iconic charm to soften expressions of digital vulnerability.
**Human annotation:** The crying panda emoji expresses sadness or amplifies empathy, making it relatable in emotional contexts.

**UMP-Net:** A man posing beside a vibrantly adorned cow, possibly during a cultural festival.
**Human annotation:** A man sits with a traditionally decorated cow.

Figure 8: Examples demonstrating UMP-Net's visual instruction-following capacity for this Instruction: Please answer me based on this image. Generate a caption of this image.

- **MG**: Text references visual content accurately (no hallucinations).

- **UH**: Calibrates or signals uncertainty; avoids overclaiming.

- **CS**: Clear, concise, well-structured outputs.

Task Descriptions:

- T1: Two-sentence definition of intelligence.

- T2: Predict null values in matrix (weights==0).

- T3: Gaussian elimination 3×3 with explanation.

- T4: Historical compare/contrast (Elizabeth I vs. Meiji).

- T5: Image captioning (COCO-like).

- T6: ScienceQA-style VQA.

In order to verify these results, we conducted a small-scale study on ScienceQA (100 samples, unclear inputs). Methods: UMP-Net vs. baselines. Measures: Aggregate rubric mark (mean 05). Findings: UMP-Net scores 4.5 ( vs. LLaMA-Excitor at 3.8), MG has increased its score by 0.8 and UH by 0.7 on ambiguous inputs Zou et al. (2024). These findings demonstrate that UMP-Net is at 4.44 ( vs. LLaMA-Excitor at 3.83), with counter-examples indicating an advantage of MG and UH in multi-modes. The MoP framework and Conformal Predictions of UMP-Net allow grounded output and justifies its benchmark improvements.

Table 20: Per-task weighted scores (0–5 scale) for UMP-Net vs. baselines.

| Task | LLaMA-Adapter | LLaMA-Excitor | LLaVA | UMP-Net |
|------|---------------|---------------|-------|---------|
| T1: Two-sentence definition of intelligence | 3.40 | 4.00 | 3.38 | 4.77 |
| T2: Predict null values in matrix | 3.57 | 3.42 | 3.08 | 4.67 |
| T3: Gaussian elimination $3\times3$ | 3.77 | 3.67 | 3.18 | 4.50 |
| T4: Historical compare/contrast | 3.94 | 4.09 | 3.73 | 4.39 |
| T5: Image captioning (COCO-like) | 3.74 | 3.90 | 3.80 | 4.20 |
| T6: ScienceQA-style VQA | 3.69 | 3.88 | 3.75 | 4.11 |

Table 21: Aggregate mean scores across tasks.

| Model | Mean Score |
|-------|------------|
| UMP-Net | 4.44 |
| LLaMA-Excitor | 3.83 |
| LLaMA-Adapter | 3.69 |
| LLaVA | 3.49 |

**Theoretical Grounding of MoP Framework.** The Mixture of Prompts framework created by UMP-Net is based on hypothesis sampling by various hypotheses, which integrates Latent Noise Prompting, Heterogeneous Clustering, and Conformal Predictions to increase the generalization and robustness of the research results Bishop (2006). Latent noise Prompting produces a wide range of prompts through Gaussian sampling, and it explores a large task space. The KNN clustering divides prompts into groups of one modality, eliminating interference. Conformal Predictions pick useful prompts that have distribution-free uncertainty, which enhances calibration and resistance toward ambiguous inputs.

To validates the MoP framework's ability to cover diverse tasks and reduce overconfidence, we conducted a diagnostic study on ScienceQA (100 samples, multi-modal visual-text questions), mirroring Section 3. Compared UMP-Net (40 prompts, K=3 clusters) to LLaMA-Excitor Zou et al. (2024) and a no-MoP variant (single prompt). Metrics:

- **Prompt Diversity**: Cosine similarity variance across prompt embeddings.

- **Calibration**: Expected Calibration Error (ECE, lower is better).

- **Accuracy**: ScienceQA accuracy (%).

Table 23 shows UMP-Net's higher prompt diversity (variance 0.85 vs. 0.62) and better calibration (ECE 0.04 vs. 0.09), driving a 3.5% accuracy gain on ambiguous inputs. This validates the MoP framework's ability to cover diverse tasks and reduce overconfidence, unlike single-prompt PEFT methods Zou et al. (2024).

**Inference Efficiency Analysis.** To analyze the inference overhead of multi-stage pipeline of UMP-Net such as latent noise prompting, clustering by KNN, conformal scoring, and attention-based fusion. our small-scale ablation experiment was to determine the cost, in terms of computation time and cost. Another addition we made is the lightweight version, UMP-Lite, that makes use of prompt selection that is cached Zhang et al. (2024). This modularity of the pipeline is more reliable but might cause a higher latency than more basic PEFT techniques Zou et al. (2024). On a subset of ScienceQA (100 samples, multi-mod visual-text questions), we tested in the same manner as in Section 3. Methods:

- **Full UMP-Net:** Complete pipeline (Section 2) with MLP prompt generation (n=40 prompts), KNN clustering (K=3), conformal scoring, and attention gating.

Table 22: Counter-examples and why UMP-Net wins.

| Task | Key Evidence | Why UMP Wins |
|------|--------------|--------------|
| T1 | Respects 2-sentence constraint; synthesizes Plato/Aristotle; explicit preference rationale. | Higher IA/RQ/FF vs. baselines that were generic or violated constraints. |
| T2 | Implements spec exactly (weights==0 → NaN), minimal code, vectorized. | Others misinterpret task or produce convoluted loops; UMP clearer & correct. |
| T3 | Partial pivoting; correct forward/back-substitution; avoids singular pitfalls. | Baselines contain structural errors (e.g., bogus rows or missing pivot checks). |
| T4 | Names concrete policies (Elizabethan Settlement; Meiji centralization); balances contrast. | More specific and sourced; baselines are surface-level or generic. |
| T5 | More granular visual nouns/relations; fewer hallucinations. | Better MG/FF; captions contain richer but grounded detail. |
| T6 | Sketched rationale consistent with visual cues; answer matches ground truth. | Slight but consistent gains in FF/RQ; clearer tie to image context. |

Table 23: Diagnostic analysis of UMP-Net's theoretical advantages on ScienceQA (100 samples).

| Method | Prompt Diversity (Var) | ECE | ScienceQA Acc (%) |
|--------|------------------------|-----|-------------------|
| UMP-Net | 0.85 | 0.04 | 88.41 |
| LLaMA-Excitor | 0.62 | 0.09 | 84.9 |
| No MoP | 0.50 | 0.12 | 82.3 |

- **UMP-Lite (Cached):** Precompute and cache cluster centroids on a calibration set (10% of training data); select nearest centroid via cosine similarity, skipping clustering and conformal steps.

- **Baseline:** LLaMA-Excitor (Zou et al., 2024), a PEFT method without uncertainty-aware components.

Metrics: Latency (ms/sample), GFLOPs, and ScienceQA accuracy (%), profiled via PyTorch's `torch.utils.benchmark`.

Table 24: Ablation experiment on the efficiency of inference, Full UMP-Net, UMP-Lite (cached prompts), and LLaMA-Excitor on ScienceQA

| Variant | Latency (ms/sample) | GFLOPs | ScienceQA Acc (%) |
|---------|---------------------|--------|-------------------|
| Full UMP-Net | 150 | 5.2 | 88.41 |
| UMP-Lite (Cached) | 95 | 3.5 | 87.8 |
| LLaMA-Excitor | 80 | 3.0 | 87.87 |

Table 24 shows Full UMP-Net's overhead (150 ms/sample, 5.2 GFLOPs) versus LLaMA-Excitor (80 ms/sample, 3.0 GFLOPs), primarily from KNN clustering (30%) and conformal scoring (25%). UMP-Lite reduces latency by 37% and compute by 33% with a minimal 0.6% accuracy drop, confirming the efficacy of cached prompt selection for frequent tasks. This supports UMP-Net's efficiency claims while addressing practical deployment concerns.

### A.1.5 Computational Efficiency Analysis

**Cost Efficiency Analysis.** Inferential latency, throughput, VRAM consumption, and FLOPs of UMP-Net were measured on a subset of ScienceQA (100 samples, multi-modal visual-text questions) with batch size (B=1, 8), text length (L=512, 1024), and image resolution (336px in vision-language, VL) to measure the computational cost of the model. We compared UMP-Net and LLaMA-Adapter, keeping the overhead of MoP. The results are presented in Table 25.

Also to isolate MoP's overhead, we profiled component timings (4090, VL, B=1, L=512, 336px). Table 26 shows the results.

As it is illustrated in results, UMP-Net's latency is modestly higher (+4.8–10.2%) than LLaMA-Adapter's, with throughput of 78.3 tokens/s and 11.0 images/s (4090, VL) and 380 tokens/s and 39.0 images/s (A100, VL). MoP components (MoPs MLP, CUE, attention gate) add only 15 ms (0.9% of 1740 ms total latency), with CLIP encoder (95 ms) and LLaMA forward (1630 ms) dominating. VRAM usage (+2.1–3.8%) and FLOPs (0.81–6.20 T) remain practical. The overhead is justified by UMP-Net's 3.5% ScienceQA accuracy gain (Table 23), supporting robust multi-modal applications.

Table 25: Inference cost (mean ± sd) for UMP-Net vs. LLaMA-Adapter. Δ% = (UMP-Net - LLaMA-Adapter) / LLaMA-Adapter.

| GPU | Task | B | L | ImgRes | Latency (ms) | Tokens/s | Images/s | VRAM (GB) | FLOPs (T) | ΔLatency% |
|------|------|---|------|--------|--------------|------------|------------|-----------|-----------|-----------|
| 4090 | Text | 1 | 512  | —      | 1300 ± 40    | 101.2 ± 3.0 | —          | 14.4      | 0.81      | +4.8%     |
| 4090 | VL   | 1 | 512  | 336    | 1740 ± 55    | 78.3 ± 2.1  | 11.0 ± 0.4 | 16.7      | 1.21      | +10.1%    |
| A100 | Text | 8 | 1024 | —      | 1930 ± 60    | 505 ± 12    | —          | 28.1      | 4.48      | +4.3%     |
| A100 | VL   | 8 | 1024 | 336    | 2600 ± 85    | 380 ± 11    | 39.0 ± 1.2 | 33.0      | 6.20      | +10.2%    |

Table 26: Component breakdown (4090, VL, B=1, L=512, 336px). Times in ms.

| Component | Time (ms) |
|-----------|-----------|
| CLIP encoder | 95 |
| MoPs MLP | 8 |
| CUE (uncertainty scoring) | 5 |
| Attention gate (softmax) | 2 |
| LLaMA forward | 1630 |

