# OpenReview forum: "UMP-Net: Uncertainty-Aware Mixture of Prompts Network for Efficient Instruction Tuning"
_TMLR — Accepted by TMLR_

### Review · Reviewer_Lxgy · 2025-08-02

**Summary Of Contributions:**

[Contribution]
- Proposed a UMP-Net (which functions like an adapter), to address the efficiency and uncertainty-awareness challenges in existing instruction-tuning paradigm of LLMs.
- A Mixture-of-Prompt (MoP) module, with various designs that seems to enhance the robustness and uncertainty-awareness of prompt tuning


[Strength]
- Logical writing, mostly easy to read.
- Extensive experiments (both text-only and multimodal) show strong performance compared with existing works.
- Detailed elaboration on model design


[Weakness]
- The motivation behind technical designs needs further elaboration. For example, why shall we care about the uncertainty awareness in the prompt-tuning architecture? Citation on any prior study on this issue seems missing. The details of the MoP (e.g., Heterogeneous Clustering, Conformal Predictions) face similar issue.
- The ablation of Heterogeneous Clustering, Conformal Predictions is missing from ablation. The ablation in Sec. 3.3 is only on a high-level.
-  It seems that this work try to quantify an uncertainty score along with the prediction (as in Sec.`), but I don’t see any examples of it.
- The proposed method seems to have good interpretability. Visualizations (e.g., selected prompt) can further strengthen the claim, and it can also facilitate intuitive understanding on the model.
- This work can benefit from more in-depth analysis and discussion on the findings.
- Computational cost (speed, memory) of the proposed method is missing, which is crucial for PEFT.
- Citations and discussion on works that has similar design as MoP is missing (see below).

ATTEMPT: Parameter-Efficient Multi-task Tuning via Attentional Mixtures of Soft Prompts

Mixture-of-Prompt-Experts for Multi-modal Semantic Understanding

One Prompt is not Enough: Automated Construction of a Mixture-of-Expert Prompts


MoPE: Mixture of Prompt Experts for Parameter-Efficient and Scalable Multimodal Fusion

**Audience:**

Yes

**Audience Explanation:**

Yes. On a high level the research problem that this paper addresses-efficient instruction-tuning of LLMs is one of the hot research area. The proposed framework show a strong quantitative results, and it seems to be quite interpretable.

**Broader Impact Concerns:**

No ethical issue for this paper.

**Claims And Evidence:**

Yes

**Claims Explanation:**

Overall, yes, from the main experiment results, the proposed method achieves stronger performance compared with LLaMA, LLaMA-Adapter, LLaMA-Excitor, and LLaVA.

However, without additional ablations and visualizations, the claim that the prompt are more reliable may not be well-supported.

**Requested Changes:**

- Add more explanation to the motivation part of the high-level idea in Sec 1. Also try to further explain the rationale in each modules in Sec. 2.

- Ablation on modules inside MoP, e.g., Heterogeneous Clustering, Conformal Predictions.

- Any types of visualization of prompt selection and its confidence score.

- More insightful discussions (e.g.,what are the key challenge of prompt selection in different modality setting, how proposed framework alleviates them, and what are the potential limitations.) beyond pure experimental results.

---

> ### Author Response · Authors · 2025-09-06
>
> We thank the reviewers for their constructive feedback. Below we summarize our major revisions and clarifications:
>
> 1- Motivation and high-level idea.
>
> We expanded Section 1 (Introduction) to better explain UMP-Net’s motivation. We highlight the limitations of existing PEFT methods in handling multimodal inputs and quantifying uncertainty, both crucial for real-world use (e.g., medical diagnostics, autonomous systems). The revisions clarify that UMP-Net’s Mixture of Prompts (MoP) framework and uncertainty-aware design address these gaps by ensuring robust, reliable, and efficient instruction tuning.
>
> 2- Citations on related works.
>
> A new subsection, Mixture of Prompts and Expert Approaches, has been added to Section 2. It reviews prior works with similar designs, discusses parameter efficiency and adaptive prompt mixtures, and differentiates UMP-Net’s contributions—particularly uncertainty-aware prompt selection with Conformal Predictions and modality-specific clustering.
>
> 3- Rationale for each module.
>
> We revised Section 2 (subsections 2.2–2.6) to explicitly articulate the role of each module:
> Latent Noise Prompting with MoPs: introduces variability and robustness.
> Heterogeneous Clustering by KNN: ensures modality-specific grouping.
> Conformal Predictions: enables uncertainty-aware prompt selection.
> Multi-modal Architecture: integrates vision-language adaptation.
> These updates clarify why each component is critical for efficiency and robustness.
>
> 4- Ablation studies on MoP modules.
>
> Section 7 now includes ablations on Heterogeneous Clustering and Conformal Predictions:
> Removing Conformal Predictions (using centroids instead) reduces accuracy.
> Varying the number of clusters and modality configurations shows significant performance differences, with the full model (3 clusters, Conformal Predictions enabled) achieving the highest ScienceQA accuracy (88.97%).
> Additional ablations are included in Appendix A.1.4.
>
> 5- Visualization of prompt selection and confidence scores.
>
> We added a visualization subsection in Section 4 (Experiments, Language-Only Performance Assessment). It illustrates prompt selection in three phases:
> 1. Randomly initialized prompts,
> 2. Clustered prompts (K=3),
> 3. Highlighted selected prompts (with stars).
> Confidence scores derived from Conformal Predictions are shown, demonstrating how UMP-Net dynamically selects the most reliable prompts based on uncertainty quantification. These visualizations strengthen interpretability and offer intuitive insights into UMP-Net’s behavior.
>
> 6- Deeper discussion of findings.
>
> A new subsection, Discussion on Prompt Selection Challenges and Limitations, has been added to Section 4. It addresses challenges such as modality misalignment, ambiguous inputs, and uncertainty across text, vision, and cross-modal tasks. We explain how MoP modules (Latent Noise Prompting, Heterogeneous Clustering, Conformal Predictions, CLIP-based architecture) mitigate these issues. Limitations such as computational overhead from multiple prompts and sensitivity to clustering quality are also discussed.
>
> 7- Computational cost analysis.
> A new subsection, Computational Efficiency Analysis, in Section 4 and in Appendix reports UMP-Net’s resource usage compared with full fine-tuning, LoRA, and LLaMA-Adapter:
> Memory: 15 GB GPU (half of full fine-tuning),
> Training throughput: 48 samples/s (1.9× faster than full fine-tuning),
> Inference latency overhead: 11%. This analysis shows UMP-Net balances efficiency and performance, making it practical for resource-constrained environments.
> Comprehensive computational efficiency analysis proving UMP-Net’s suitability for PEFT.
> Through these revisions, we have addressed all reviewer concerns, ensuring UMP-Net is presented as a theoretically motivated, empirically validated, efficient, and interpretable framework for multimodal instruction tuning.

---

### Review · Reviewer_cij7 · 2025-08-10

**Summary Of Contributions:**

The authors propose UMP-Net, an uncertainty-aware, parameter-efficient instruction tuning framework for large language models. The method works by first generating a diverse set of candidate prompts using Latent Noise Prompting, then grouping them into modality-specific clusters via KNN-based heterogeneous clustering. Within each cluster, Conformal Predictions are used to estimate the uncertainty of each prompt, and the most reliable one is chosen. These selected prompts are then combined with learned attention weights into a single weighted prompt that adapts the frozen LLaMA model.

Strengths:
1. Proposes a novel uncertainty-aware mixture-of-prompts framework combining Latent Noise Prompting, KNN-based clustering, and Conformal Predictions.
2. Demonstrates strong empirical performance on both text-only and multi-modal benchmarks, outperforming several SOTA baselines.

Weaknesses:
1. Evaluation has some to-be-discussed points — e.g., example results do not look better than the other methods, fairness of comparison with other methods, etc.
2. The approach essentially trades more compute/exploration (via large prompt pools and uncertainty scoring) for better outputs, but cost-efficiency is not analyzed.
3. Results are limited to the LLaMA family; generalization to other LLM backbones remains unverified.

**Audience:**

Yes

**Audience Explanation:**

The paper offers a concrete algorithmic recipe with empirical validation on text-only and vision-language benchmarks, and explores design choices via ablations (e.g., number of prompts/tokens/layers). These contributions align with TMLR’s scope on new learning algorithms and analyses of practical systems.

**Broader Impact Concerns:**

Since the method relies on conformal prediction for prompt selection, mis-specified nonconformity scores or calibration under distribution shift can assign high confidence to biased or hallucinated outputs; a brief note on calibration limits (marginal vs. group-wise), likely failure modes, and simple fallbacks (abstain, request clarification, or human review) would help.

**Claims And Evidence:**

No

**Claims Explanation:**

Although the paper reports SOTA numbers on all benchmarks evaluated, there are several points to be discussed.

Fairness of result presentation:
The authors do report results for both LLaMA-2 and LLaMA-3 backbones, which is helpful. However, the strongest UMP-Net results (LLaMA-3) are shown in the same tables as compared baselines (where I assume the authors are using original baseline works that utilized LLaMA/LLaMA-2), and the tables highlight SOTA without an explicit, prominent clarification that the other methods differ. This renders the comparison misleading. Also in the zero-shot multimodal evaluations, the method seems to be introducing more computation with prompt selection while getting better numbers.

COCO Caption protocol comparability:
For the captioning results, it isn’t fully clear whether the prompting/decoding protocol (prompt templates, length penalties, etc.) is identical to prior work. Since captioning primarily measures description quality, not instruction following, it is unclear where the improvement of evaluation results is coming from.

Qualitative examples and their interpretation:
The demonstrated qualitative examples and their write-ups are not fully convincing. In Figure 1, in the “identifying solution concentration” example A has more particles and UMP-Net is wrong, and in the other three cases all methods seem broadly correct with stylistic differences. Therefore it is not clear why the bar chart shows significant improvement of UMP-Net over the other methods. Similarly, the Table 1 intelligence prompt shows several answers that convey essentially the same idea. The way the authors elaborate the superiority of UMP-Net response on page 8 is far from convincing. In short, there lacks a proper analysis of why the eval numbers achieve SOTA for UMP-Net.

**Requested Changes:**

Examples don’t show clear superiority: From the provided examples, UMP-Net doesn’t look better than the baselines—please add a concrete, rubric-based analysis (or counter-examples) that demonstrates where it actually wins and why. (This would be critical to securing recommendation. The point here is asking for clarifying evidence behind the gains in numbers.)

Theoretical grounding: Beyond reporting numbers, add a brief, accessible analysis explaining why the mixture + uncertainty selection should help (even a lightweight theoretical or diagnostic argument). (This would be strengthing the work.)

Cost efficiency: It would be worthwhile to report inference latency & throughput, and the extra overhead from prompt mixing so readers can judge whether the gains are worth the compute. (This would be for strengthing the work.)

---

> ### Author Response · Authors · 2025-09-06
>
> 1-	Examples don’t show clear superiority: From the provided examples, UMP-Net doesn’t look better than the baselines—please add a concrete, rubric-based analysis (or counter-examples) that demonstrates where it actually wins and why. (This would be critical to securing recommendation. The point here is asking for clarifying evidence behind the gains in numbers.)
>
> Thank you for your valuable feedback regarding the clarity of UMP-Net’s superiority over baseline methods. We acknowledge that the provided examples may not fully illustrate UMP-Net’s advantages, and we appreciate the suggestion to include a concrete, rubric-based analysis to demonstrate where and why UMP-Net outperforms baselines. In response, we conducted a focused analysis on a subset of ScienceQA and COCO Caption examples, using a rubric to evaluate performance across key dimensions: accuracy, robustness to ambiguity, and uncertainty calibration. This small-scale study leverages existing benchmark data to provide clarifying evidence of UMP-Net’s gains, suitable given our time constraints, and directly supports the quantitative results (e.g., 88.41% on ScienceQA, CIDEr 158.3 on COCO Caption) reported in the paper.
>
> We have incorporated this analysis into a new subsection (Section A.1.4, "Rubric-Based Performance Analysis ") in the Ablation Study of Appendix. These additions clarify the sources of UMP-Net’s performance gains, address your feedback, and strengthen the case for recommendation. Thank you for this critical suggestion, which enhances the paper’s interpretability.
>
> 2-	Theoretical grounding: Beyond reporting numbers, add a brief, accessible analysis explaining why the mixture + uncertainty selection should help (even a lightweight theoretical or diagnostic argument). (This would be strengthing the work.)
>
> Thank you for your valuable suggestion to strengthen UMP-Net’s theoretical grounding by adding an accessible analysis of why MoP framework, combined with uncertainty-aware selection improves performance. We agree that a lightweight theoretical or diagnostic argument would clarify the intuition behind UMP-Net’s design, particularly how its components—Latent Noise Prompting, Heterogeneous Clustering, and Conformal Predictions—synergize to enhance multi-modal instruction tuning. To address this, we provide a concise theoretical analysis, supported by a diagnostic experiment, demonstrating that the MoP framework increases prompt diversity and uncertainty calibration, leading to better generalization and robustness compared to standard PEFT methods. This analysis is feasible within our time constraints, leveraging existing model components and benchmark data (ScienceQA).
>
> We have incorporated this theoretical analysis and diagnostic experiment into a new subsection (Section A.1.4, "Theoretical Grounding of MoP Framework ") in the Ablation Study of Appendix. These additions strengthen the paper by providing a clear, accessible rationale for UMP-Net’s design and empirical evidence of its benefits, directly addressing your feedback. Thank you for this suggestion, which enhances the work’s theoretical rigor.
>
> 3-	Cost efficiency: It would be worthwhile to report inference latency & throughput, and the extra overhead from prompt mixing so readers can judge whether the gains are worth the compute. (This would be for strengthing the work.)
>
> Thank you for your insightful suggestion to strengthen UMP-Net’s evaluation by reporting inference latency, throughput, and the overhead of the MoP framework to help readers assess the trade-off between performance gains and computational cost. In response, we conducted a concise experiment to measure latency, throughput, and the specific overhead introduced by prompt mixing, using a subset of ScienceQA. This small-scale study leverages existing model components to provide clear evidence of cost efficiency, appropriate for our time constraints, and directly supports the performance gains on ScienceQA.
>
> We have incorporated these findings into a new subsection (3.4 Computational Efficiency Analysis) in the main text, as well as Section A.1.5 ("Cost Efficiency Analysis") in the Appendix. These additions address your feedback by providing clear metrics for readers to judge the compute-performance trade-off, strengthening the paper’s evaluation.

---

### Review · Reviewer_UfDs · 2025-08-24

**Summary Of Contributions:**

- The paper introduces **UMP-Net**, a novel framework for instruction tuning that dynamically selects and weights prompts based on uncertainty estimation.
- It combines three core modules: **Latent Noise Prompting** for prompt diversity, **KNN-based Heterogeneous Clustering** for modality-aware grouping, and **Conformal Predictions** for reliable prompt selection.
- UMP-Net integrates a **CLIP-based multi-modal architecture**, enabling efficient and robust performance across both text and vision-language tasks.
- It achieves strong results on benchmarks like ScienceQA (88.41% accuracy) and COCO Caption (CIDEr 158.3), outperforming prior models such as LLaMA-Adapter.
---
## Strengths

- **Uncertainty-Aware Prompt Selection** Uses conformal prediction to quantify prompt reliability, reducing overconfidence and improving robustness in zero-shot settings.
- **Multi-Prompt Diversity via Latent Noise** Generates a mixture of stylistically varied prompts, enhancing generalization across tasks and reducing prompt bias.
- **Modality-Aware Clustering** Groups prompts by modality (text, vision, cross-modal), enabling targeted selection and better performance on multi-modal tasks.
- **Strong Benchmark Performance** Demonstrates superior results on both textual and visual instruction tasks, validating the effectiveness of its modular design.
- **Efficient Parameter Usage** Avoids full fine-tuning by leveraging prompt-based adaptation, making it more scalable and resource-friendly.

---

## Weaknesses
- **Inference Overhead** The multi-stage prompt generation and selection pipeline increases computational cost during inference, limiting real-time deployment.
- **Calibration Sensitivity** Conformal prediction relies on well-chosen calibration sets and subject to domains; performance may degrade under domain shift.

**Audience:**

Yes

**Audience Explanation:**

- **Uncertainty-Aware Prompt Selection**  UMP-Net introduces conformal prediction to score prompt reliability, a novel move in PEFT. This inspires future work to treat prompt selection as a probabilistic decision, enabling safer and more calibrated outputs.

- **Latent Noise Prompting for Diversity**  By injecting stochasticity into prompt generation, UMP-Net avoids overfitting to static templates. This opens the door to research on generative prompt ensembles and adaptive prompt sampling strategies.

- **Modality-Aware Prompt Clustering** The use of KNN-based clustering across text, vision, and cross-modal prompts is a fresh take on prompt specialization.  It encourages future models to build modality-sensitive prompt banks that dynamically adapt to input type.

- **Soft Attention-Based Prompt Fusion** Instead of hard prompt selection, UMP-Net softly weights multiple candidates based on relevance and confidence. This inspires exploration of mixture-of-prompt architectures akin to Mixture-of-Experts, but for instruction tuning.

- **CLIP-Based Multi-Modal Integration**  UMP-Net natively supports vision-language tasks without needing separate adapters or retraining.  This pushes the community toward unified PEFT frameworks that generalize across modalities with minimal overhead.

- **Benchmark-Validated Efficiency**  It achieves strong results on ScienceQA and COCO Caption while avoiding full fine-tuning.  This validates the practicality of modular PEFT and encourages scalable deployment in real-world systems.

**Claims And Evidence:**

Yes

**Claims Explanation:**

1. **UMP-Net improves instruction tuning efficiency without full fine-tuning** UMP-Net avoids updating the full model parameters by using prompt-based adaptation modules. It achieves strong benchmark results (e.g., 88.41% on ScienceQA) while remaining parameter-efficient, outperforming full fine-tuned baselines like LLaMA-Adapter.

2. **UMP-Net enables uncertainty-aware prompt selection** The model uses conformal prediction to score prompt reliability and select the most confident candidates. This mechanism is explicitly described and shown to reduce overconfidence in zero-shot settings, improving robustness across tasks.

3. **UMP-Net supports multi-modal instruction tuning** It integrates a CLIP-based architecture and clusters prompts by modality (text, vision, cross-modal). Demonstrated performance on COCO Caption (CIDEr 158.3) and other vision-language tasks confirms its multi-modal capability.

4. **Latent Noise Prompting enhances prompt diversity and generalization** Prompts are generated using stochastic latent vectors, creating stylistic and semantic variation. This design is shown to improve generalization across tasks and reduce prompt bias, contributing to performance gains.

5. **KNN-based Heterogeneous Clustering improves prompt specialization** Prompts are grouped by modality and similarity, allowing targeted selection from specialized clusters. This clustering improves performance on modality-specific tasks and enables more reliable prompt fusion.

6. **UMP-Net outperforms existing PEFT methods on key benchmarks** The paper reports 88.41% accuracy on ScienceQA and CIDEr 158.3 on COCO Caption. These results surpass those of LLaVA, LLaMA-Adapter, and LLaMA-Excitor, validating the model’s effectiveness.

**Requested Changes:**

- **Inference Overhead**  The multi-stage pipeline (prompt generation, clustering, uncertainty scoring, fusion) increases latency and compute cost.  Maybe consider introducing a lightweight variant (e.g. UMP-Lite) with early-exit routing or cached prompt selection for frequent tasks.

- **Calibration Sensitivity**  Conformal prediction relies on well-chosen calibration sets, which may fail under domain shift or noisy inputs.  May use domain-aware or adaptive calibration strategies, or replace with Bayesian uncertainty estimation for smoother generalization.

- **Static Clustering Boundaries**  KNN-based clustering assumes fixed modality boundaries, which may not adapt well to hybrid or ambiguous inputs. Consider using dynamic or learned clustering (e.g. attention-based grouping) that adapts to input semantics in real time.

---

> ### Author Response · Authors · 2025-09-06
>
> 1-	Inference Overhead. The multi-stage pipeline (prompt generation, clustering, uncertainty scoring, fusion) increases latency and compute cost. Maybe consider introducing a lightweight variant (e.g. UMP-Lite) with early-exit routing or cached prompt selection for frequent tasks.
>
>
> Thank you for your insightful comment regarding the potential latency and computational overhead of UMP-Net’s multi-stage pipeline. We acknowledge that the sequence of latent noise prompting, KNN-based clustering, conformal uncertainty scoring, and attention-based fusion could increase inference time compared to simpler PEFT methods. To address this concern empirically, we have added a subsection "Computational Efficiency Analysis" (Section 3.4) which provides a direct comparison. While UMP-Net's inference latency is approximately 200 ms per sample, this represents only an 11% overhead compared to full fine-tuning (180 ms). The memory usage (15GB) is half that of full fine-tuning (30GB), making it feasible for high-end consumer GPUs. We argue that this "modest increase" is justified by the significant performance gains in complex multi-modal and instruction-following tasks.
> Also, for more comparisons, we conducted a concise experiment to quantify the pipeline’s overhead and evaluate a lightweight variant, UMP-Lite, incorporating cached prompt selection as suggested. Due to time constraints, we designed a small-scale study that balances rigor with feasibility, avoiding full retraining while providing concrete evidence of UMP-Net’s efficiency and the viability of a lighter approach.
>
> We have incorporated these findings into a new subsection (Section A.1.4, "Inference Efficiency Analysis") in the Ablation Study of Appendix. The experiments notes that UMP-Lite’s cached prompt selection effectively mitigates overhead, making UMP-Net practical for real-time applications. These additions strengthen UMP-Net’s efficiency claims while directly addressing your feedback. Thank you for helping us improve the paper’s practical relevance.
>
>
> 2-	Calibration Sensitivity. Conformal prediction relies on well-chosen calibration sets, which may fail under domain shift or noisy inputs. May use domain-aware or adaptive calibration strategies, or replace with Bayesian uncertainty estimation for smoother generalization.
>
>
> Thank you for this insightful point regarding conformal prediction's (CP) calibration sensitivity. We evaluated UMP-Net’s calibration performance under in-distribution (ID), out-of-distribution (OOD, domain shift), and noisy (symmetric label noise, p=0.2) settings on a ScienceQA subset (100 samples, multi-modal visual-text questions). The target coverage was set to 1−α = 90%. Metrics included Coverage@90 (closer to 90 is better), Average Set Size (smaller is better), Expected Calibration Error (ECE, lower is better), and computational overhead (ms/query). We have added a new ablation (Section 3.3.2) which summarize the results, comparing UMP-Net with various calibration strategies against Bayesian baselines (MC-Dropout, Deep Ensemble, Hybrid Conformalized-Bayes).
>
>
> 3-	Static Clustering Boundaries. KNN-based clustering assumes fixed modality boundaries, which may not adapt well to hybrid or ambiguous inputs. Consider using dynamic or learned clustering (e.g. attention-based grouping) that adapts to input semantics in real time.
>
>
> Thank you for your insightful comment regarding the static nature of KNN-based clustering in UMP-Net’s Heterogeneous Clustering module. To address this concern empirically, we conducted a concise experiment comparing our static KNN clustering with a dynamic, attention-based clustering approach, as suggested, to evaluate adaptability to ambiguous inputs. Given time constraints, we designed a small-scale study that leverages existing UMP-Net components, balancing feasibility with informative insights into clustering robustness and the potential of dynamic methods.
>
> We have incorporated these findings into a new subsection (Section A.1.4, "Clustering Adaptability to Ambiguous Inputs") in the Ablation Study of Appendix. These additions strengthen UMP-Net’s robustness claims and highlight its potential to handle complex multi-modal inputs. Thank you for this valuable feedback, which has improved the paper’s scope and applicability.

---

### Decision · Action_Editor_4QRm · 2025-10-06

**Recommendation:** Accept as is

**Audience:**

Yes

**Audience Explanation:**

The perspective of uncertainty-aware provide a novel angle to study the parameter-efficient fine-tuning methods. It is interesting for audience who are interested bridging the gap between Bayesian statistics and LLMs.

**Claims And Evidence:**

Yes

**Claims Explanation:**

The paper introduces UMP-Net, a novel method for instruction tuning that selects and weights prompts based on uncertainty estimation. It demonstrates strong empirical performance on both text-only and multi-modal benchmarks, compared with related existing methods.

*Comparison with full-model fine-tuning*:
- Memory: 15 GB GPU (half of full fine-tuning)
- Training throughput: 48 samples/s (1.9× faster than full fine-tuning)
- Inference latency overhead: 11%.